# Exploiting Spatial Separability for Deep Learning Multichannel Speech Enhancement with an Align-and-Filter Network

## Abstract

Multichannel speech enhancement (SE) systems separate the target speech from background noise by performing spatial and spectral filtering. The development of multichannel SE has a long history in the signal processing field, where one crucial step is to exploit spatial separability of sound sources by aligning the microphone signals in response to the target speech source prior to further filtering processes. This is similar to the human listening behavior of facing toward the speaker for better perception of the speech. However, most existing deep learning-based multichannel SE works have yet to effectively incorporate or emphasize this spatial alignment aspect in the network design; some of them rely on integrating conventional model-based beamformer units to extract useful spatial features implicitly while others just let the network figure everything out by itself. However, the beamformer operation could be computationally expensive and numerically unstable when trained with the network while without it the model lacks guidance on learning meaningful spatial features. In this paper, we highlight this important but often overlooked step in deep learning-based multichannel SE, i.e., signal alignment, by introducing an Align-and-Filter network (AFnet) featuring a two-stage sequential masking design. The AFnet aims at estimating two sets of masks, the alignment masks and filtering masks, to carry out temporal alignment and spectral filtering processes. During training, we propose to supervise the learning of alignment masks by predicting the relative transfer functions (RTFs) of various speech source locations followed by learning the filtering masks for signal reconstruction. During inference, the AFnet sequentially multiplies the estimated alignment and filtering masks with the microphone signals, performing the "align-then-filter" process similar to the human listening behavior. Due to the incorporation of RTF supervision, the AFnet explicitly learns interpretable spatial features without integrating traditional beamformer operations.

## 1 Introduction

Speech enhancement (SE) systems can be categorized into *single-channel* (single microphone) and *multichannel* (multiple microphones) schemes. An important aspect of multichannel SE against single-channel SE is the *exploitation of spatial separability*, as known as *spatial filtering* or *beamforming*, enabled by the *difference* between the amplitudes and times of arrival of the received microphone signals due to the different acoustic paths the sound waveform travels to the microphones. In many signal processing beamforming methods (Gannot et al., 2001; Cohen, 2004; Krueger et al., 2010; Koldovskỳ et al., 2015), a key step is to *align* the microphone signals in response to the target signal source before any further filtering processes. This step, by *steering* the array toward the location of the target signal, aims to compensate for the difference of the amplitudes and time delays (or correspondingly the magnitudes and phases in the frequency domain) of the microphone signals with respect to the target source. Ideally, after the alignment step, each microphone should contain the same target speech component with no difference in amplitude and time delay (or magnitude and phase). For a linear array in the far-filed, anechoic setting, perfectly steering the microphone array makes it as if the target signal comes from the broadside, which renders the speech extraction task easier in the later filtering stage. Such process is similar to the human listening behavior of facing toward the speaker for better perception of the speech. Thus, an efficient SE system can first align its microphone signals in response to the target speech, followed by spectral processing for fine-tuning

and enhancement. Surprisingly, though well-known in signal processing approaches, there are only few deep learning SE systems designed mainly based on this observation. Instead, several recent works (Wang et al., 2020; 2021) have been utilizing conventional beamformer units such as the the minimum-variance-distortionless-response (MVDR) beamformer (Capon, 1969) to extract spatial characteristics implicitly in deep learning methods. However, matrix inversion or eigendecomposition are often required which in turn increase the computation burden and numerical instability.

In this paper, we revisit this important but often overlooked alignment aspect from conventional signal processing algorithms and recognize its importance for efficient deep learning multichannel SE network design that requires no intermediate beamformer units such as MVDR to extract meaningful spatial features. Specifically, we propose the Align-and-Filter network (AFnet) shown in Figure 1 for exploiting spatial separability within multichannel data with the following main contributions:

- The AFnet features a two-stage sequential masking design, i.e., Align Net and Filter Net, where two sets of masks, alignment and filtering masks, are estimated and multiplied with the microphone signals to perform the "align-then-filter" process *mimicking the human listening behavior*.

- During the training stage, we propose to supervise the learning of the alignment masks by estimating the *relative transfer functions (RTFs)* (Gannot et al., 2001; Cohen, 2004) for speech sources coming from various locations, prior to learning the filtering masks for final enhancement.

- During inference, the AFnet is able to first align the microphone signals with respect to a speech source coming from an unknown direction due to supervised learning of alignment masks. Subsequently, the model performs filtering on the roughly aligned signals to achieve denoising.

- It is demonstrated that the RTF supervision incorporated with the sequential masking mechanism is the key to effectively learn useful, interpretable spatial characteristics. On situations where the target speech may come from arbitrary positions, the "align-then-filter" mechanism consistently improves the SE performance by more efficiently exploiting spatial separability of sound sources.

## 2 RELATED WORK

Multichannel SE has been a well known topic in signal processing for decades. Aside from leveraging spectral characteristics, multichannel SE can exploit positional information to perform *spatial filtering*, or *beamforming*, that allows extracting the target speech from noise based on spatial separability. Conventional beamforming approaches rely on the so-called "steering vector" which carries positional information about the target speech (Doclo et al., 2015; Trees, 2004), e.g., the MVDR beamformer (Capon, 1969) and its variants (Frost, 1972; Griffiths & Jim, 1982). It is an important step in beamforming that the microphone array is steered toward the target signal location prior to further filtering processes. To this end, certain knowledge about the acoustic paths between the target signal and the microphones have to be known. Many signal processing-based multichannel SE systems utilize the ratio of the acoustic transfer functions, i.e., the *relative transfer function (RTF)*, that represents the coupling between sensors in response to a desired source (Gannot et al., 2001; Cohen, 2004; Krueger et al., 2010; Koldovský et al., 2015) for improving the denoising process.

In the past decade, deep learning approaches have remarkably changed the way of developing SE systems. Along with the success of deep neural networks (DNNs) on single-channel SE (Lu et al., 2013; Williamson et al., 2015; Pascual et al., 2017; Luo & Mesgarani, 2019; Kim et al., 2020; Zheng et al., 2021; Hu et al., 2020), several multichannel SE systems have also been proposed, e.g., Erdogan et al. (2016); Variani et al. (2016); Sainath et al. (2017); Wang & Wang (2018); Bu et al. (2019); Koyama & Raj (2019); Luo et al. (2020); Tolooshams et al. (2020); Koyama & Raj (2020); Wang et al. (2021); Zhang et al. (2021); Kim et al. (2022); Li et al. (2022). However, the utilization of such important RTF information is often overlooked in the model design of DNN-based multichannel SE. Although several SE works have incorporated RTFs (Wang & Wang, 2018; Zhang et al., 2021) the RTF estimation is mostly used as an intermediate step to assist the MVDR beamformer only. We postulate that the overlook may be due to the lack of sufficient spatial variety of the speech sources in popular datasets such as CHiME-3 Barker et al. (2015), where the benefit of utilizing RTFs could be only marginal. However, many practical situations can have the target speech coming from arbitrary directions and thus a deep dive into the RTF spatial alignment aspect is still of great importance.

## 3 PROPOSED METHOD

We consider an acoustic scenario with one desired speech source and several interfering noise signals in a reverberant environment. The SE system will be developed in the time-frequency domain using

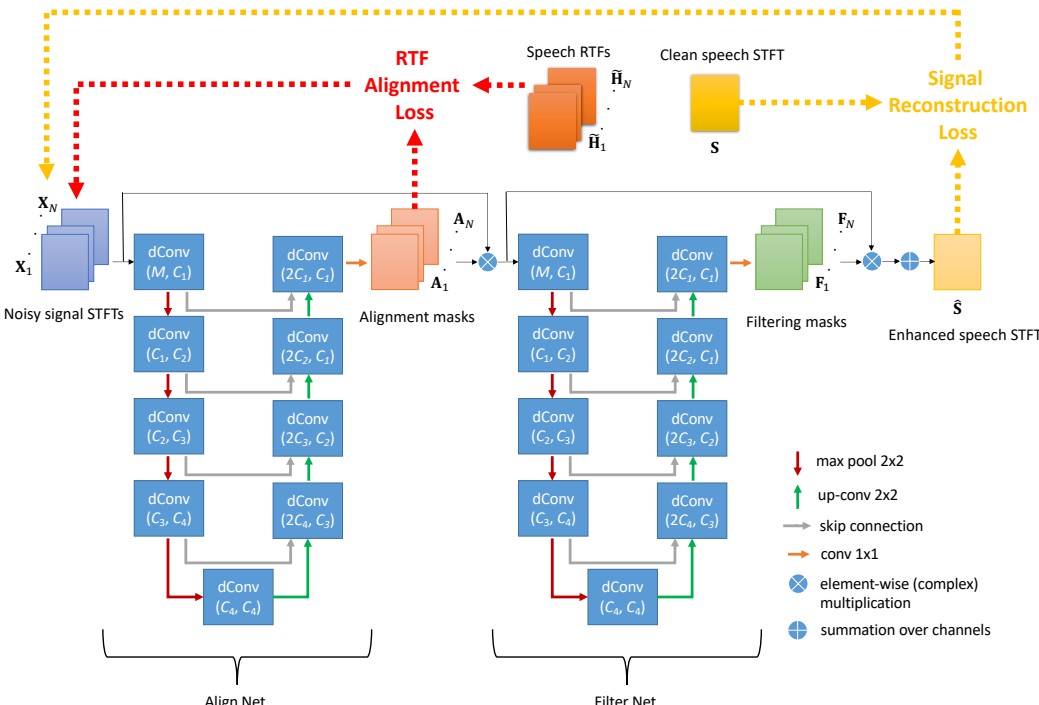

Figure 1: The proposed AFnet for exploiting spatial separability for multichannel SE. During training, the Align Net estimates the alignment masks by predicting the speech RTFs. The Filter Net estimates the filtering masks to reconstruct the clean speech by linearly combining the aligned signals after the Align Net. The RTF supervision and the two-stage sequential masking design contribute to the improved multichannel SE performance.

the short-time Fourier transform (STFT) (Parchami et al., 2016) where the time-domain waveforms are transformed to complex-valued STFT representations. Let $f, t$ stand for the frequency index and time frame index (with a total of $F$ frequency bins and $T$ time frames), we consider an additive noise model where the $i$-th microphone noisy signal STFT $\mathbf{X}_i \in \mathbb{C}^{F \times T}$ of an $N$-element local microphone array can be written as (Doclo et al., 2015):

$$X_i(f,t) = \underbrace{H_i(f,t)S_0(f,t)}_{S_i(f,t)} + V_i(f,t), \tag{1}$$

where $S_i(f,t)$ is the speech component received by microphone $i$, $H_i(f,t)$ is the (potentially time-varying) acoustic transfer function between microphone $i$ and the location of the far-field speech source signal $S_0(f,t)$, and $V_i(f,t)$ is the noise component captured by microphone $i$. Typically, the goal of multichannel SE is to recover the speech component $\mathbf{S} = \mathbf{S}_r \in \mathbb{C}^{F \times T}$ of a selected reference microphone $r \in \{1, \ldots, N\}$ given the noisy microphone signals $\mathbf{X}_1, \ldots, \mathbf{X}_N$. Multichannel SE systems typically perform the "filter-and-sum" operation, or more generally known as "beamforming" – linearly combining the signals of different microphones to extract the target signal from background noise. In the STFT domain, the estimation process can be expressed as:

$$\hat{\mathbf{S}} = \sum_{i=1}^{N} \mathbf{W}_i \odot \mathbf{X}_i, \tag{2}$$

where $\mathbf{W}_i \in \mathbb{C}^{F \times T}$ is the corresponding set of filter weights for microphone $i$, $\hat{\mathbf{S}}$ is the enhanced signal, and $\odot$ denotes element-wise complex multiplication.

### 3.1 TWO-STAGE ALIGN-AND-FILTER FRAMEWORK WITH SEQUENTIAL MASKING

Inspired by the alignment concept in signal processing-based algorithms described in Section 1 and Appendix A, we introduce a two-stage Align-and-Filter framework for deep learning multichannel

SE in Figure 1, where two modules are utilized to carry out alignment and filtering purposes respectively. The first module (Align Net) aims at estimating a set *alignment masks* for steering the input microphone signals toward the target speech location The second module (Filter Net) focuses on cleaning up the interference in the aligned signals by estimating another set of *filtering masks* and linearly combines all channel outputs. Overall, the enhancement process can be expressed as:

$$\hat{\mathbf{S}} = \sum_{i=1}^{N} (\mathbf{F}_i \odot \mathbf{A}_i) \odot \mathbf{X}_i, \tag{3}$$

where $\mathbf{A}_i \in \mathbb{C}^{F \times T}$ is the alignment mask and $\mathbf{F}_i \in \mathbb{C}^{F \times T}$ is the filtering mask of the $i$-th microphone. This corresponds to having $\mathbf{W}_i = \mathbf{F}_i \odot \mathbf{A}_i$ for the equivalent beamforming filter weights applied on $\mathbf{X}_i$ in (2). Such a sequential masking scheme also introduces signal level skip connections on top of the two modules for improved optimization.

### 3.2 RTF-AWARE TRAINING

The spatial relationship between the far-field target source and the microphones along with the acoustic environment characteristics determine the inter-channel level difference (ILD) and inter-channel time difference (ITD) of the microphone signals, which are important cues for sound source localization. In the STFT domain, the ILD and ITD correspond to the magnitude and phase of the RTF respectively, which is the ratio between the acoustic transfer functions of two microphones with regards to the sound source (Li et al., 2016). Formally, we can express the RTFs $\tilde{\mathbf{H}}_i \in \mathbb{C}^{F \times T}$ as:

$$\tilde{H}_i(f,t) \triangleq \frac{H_r(f,t)}{H_i(f,t)} = \frac{H_r(f,t)S_0(f,t)}{H_i(f,t)S_0(f,t)} = \frac{S_r(f,t)}{S_i(f,t)}, \tag{4}$$

$\forall f, t$, where $H_r(f,t)$ denotes the acoustic transfer function of the selected reference microphone $r \in \{1, \ldots, N\}$. From (4) we can see that the RTFs can be inferred from the ratios of the clean microphone signals which are often available in supervised learning SE schemes, and thus can be collected without knowing or simulating the noise. Note that the RTF information is only needed for training and is not required during inference.

Element-wise multiplying the RTFs $\tilde{\mathbf{H}}_i$ with input $\mathbf{X}_i$ we have $\mathbf{Z}_i = \tilde{\mathbf{H}}_i \odot \mathbf{X}_i \in \mathbb{C}^{F \times T}$ where:

$$Z_i(f,t) = \tilde{H}_i(f,t)X_i(f,t) = \frac{H_r(f,t)}{H_i(f,t)}[H_i(f,t)S_0(f,t) + V_i(f,t)] = S_r(f,t) + \tilde{V}_i(f,t), \tag{5}$$

$\forall f, t$, where $S_r(f,t)$ is the speech component of the reference microphone and $\tilde{V}_i(f,t) = [H_r(f,t)/H_i(f,t)]V_i(f,t)$ is the noise component. It can be seen that multiplying the RTFs with the respective input STFT leads to aligning all the microphones with respect to the clean speech component at the reference channel as each $Z_i(f,t)$ contains $S_r(f,t)$. In the case of far-field, anechoic setting, this means the microphone array has steered to the direction of the signal source. For the case of linear arrays, it can be viewed as the signal is coming from the broadside after steered.

We incorporate RTFs during the model learning phase via an RTF-aware training protocol consisting of two phases: First adjusting only the Align Net parameters so that the alignment mask $\mathbf{A}_i$ approximates the RTFs $\tilde{\mathbf{H}}_i$ by minimizing the following RTF alignment loss:

$$\mathcal{L}_{\text{rtf}} = \frac{1}{NFT} \sum_{i,f,t} |A_i(f,t) - \tilde{H}_i(f,t)|^2, \tag{6}$$

where we have chosen to use the mean squared error (MSE) loss. Note that truncation of the magnitude of the ground-truth RTFs $\tilde{H}_i(f,t)$ to a maximum value of, e.g., 10, can be applied for avoiding training instability, and the alignment mask corresponding to the reference microphone is always all ones and is possible not to be estimated. Optimizing for this loss function guides the network to learn to steer the microphone signals toward the target speech, making the model aware of the spatial information of the target source. The second phase performs parameter learning of the whole model (i.e., jointly adjusting both Align Net and Filter Net parameters) to reconstruct the clean speech. To this end, we minimize the signal reconstruction loss:

$$\mathcal{L}_{\text{rec}} = \sum_{f,t} (1-\beta)(|\hat{S}(f,t)|^c - |S(f,t)|^c)^2 + \beta ||\hat{S}(f,t)|^c e^{j\angle \hat{S}(f,t)} - |S(f,t)|^c e^{j\angle S(f,t)}|^2, \tag{7}$$

where we have chosen to use the combined power-law compressed MSE loss first appeared in (Wilson et al., 2018). In this work we use $\beta = 0.3$ and $c = 0.3$ as suggested by a consolidated study in (Braun & Tashev, 2021).

### 3.3 AFNET BASED ON COMPLEX W-NET ARCHITECTURE

We introduce the AFnet (Figure 1) for multichannel SE based on implementing a W-Net (Xia & Kulis, 2017) architecture suitable for realizing the two-stage framework. For the SE application here, extension to complex operations can be more suitable when working in the STFT domain. Choi et al. (2018) have proposed to utilize complex-valued U-Net for the single-channel SE task. We follow this idea and extend the W-Net by incorporating complex operations for multichannel SE. In the current work, main modifications to the W-Net include: Convolutional layers of W-Net are all replaced to complex convolutional layers. For the activation function, complex leaky ReLU, i.e., an activation function which applies leaky ReLU on both real and imaginary values, are utilized. The number of feature maps in each layer are also modified. Note that in Figure 1, "dConv ($C_{in}$, $C_{out}$)" stands for "double Convolutions" with $C_{in}$ input channels and $C_{out}$ output channels. Each dConv unit consists of two stacks of "(complex) 3×3 convolution→batch normalization→leaky ReLU." The convolution layer of the first stack takes $C_{in}$ channels and outputs $C_{out}$ channels; the convolution layer of the second stack takes $C_{out}$ channels and outputs $C_{out}$ channels.

### 3.4 DIFFERENTIATION FROM EXISTING METHODS

Existing works such as (Wang et al., 2021; Zhang et al., 2021) leverage the conventional beamformer modules for incorporating spatial features implicitly. However, matrix inversion or eigendecomposition are often required and additional network modules might otherwise be needed to replace such operations. Other works (Koyama & Raj, 2019; 2020; Li et al., 2022) concatenate two DNN modules and jointly train both toward the enhanced speech. However, without utilizing any beamforming units within the model it is hard to interpret if the network actually learns to exploit spatial features. Differently, our method introduces a simple skip connection of alignment masks trained via supervised learning of the RTFs to explicitly incorporate meaningful spatial characteristics.

## 4 EXPERIMENTS

### 4.1 DATASETS AND EXPERIMENTAL SETUP

#### 4.1.1 PRIMARY DATASET

**Speech corpus:** We leverage the AVSpeech dataset which is a large collection of video clips of single speakers talking with no audio background interference introduced in Ephrat et al. (2018). The dataset is based on public instructional YouTube videos, from which short, 3-10 second clips were automatically extracted, where the only audible sound in the soundtrack belongs to a single speaking person. We downloaded 8308 clips from the training set and another 1199 clips from the test set for our experiments, using only the audio portion (converted to 16 kHz .wav files).

**Room impulse response (RIR) data:** We mix the speech files with noise files of several types also downloaded from YouTube and consider different reverberation conditions for generating the noisy data using simulated impulse responses via Pyroomacoustics (Scheibler et al., 2018), which is a Python-based acoustic simulator for generating different acoustic scenes. We create a room of size $8 \times 8 \times 3$ (length×width×height in meters). Reverberation time is randomly chosen from {0.16, 0.32, 0.48, 0.64} second. The simulating approach for estimating RIRs is based on the image source method. Speech and noise signals are randomly positioned at a distance of {1, 2, 3, 4} meters from the microphone array with arbitrary directions of arrival. The microphones form a 2-by-4 planar array placed vertically at the center of the room, and the spacing between two adjacent microphones is 5 cm. 4 types of commonly seen interference (blender, vacuum, washer, baby cry) are used for training and another 4 types (dog barking, kids playing sound, hair dryer, food sizzling) are used for testing. Each type of noise is multiplied with a scale randomly chosen from {0, 0.5, 1, 1.5, 2} before added up, and the combined noise is then added to the clean speech according to a specific SNR level randomly selected from {-10, -6, -3, 0, 3, 6, 10} dB.

#### 4.1.2 AUXILIARY DATASET:

The CHiME-3 dataset (Barker et al., 2015), made available as part of a speech separation and recognition challenge, is used for training and evaluating SE performance. The dataset is a 6-channel microphone recording of talkers speaking in a noisy environment, sampled at 16 kHz. It consists of 7,138 and 1,320 simulated utterances with an average length of 3 seconds for training and testing, respectively. We take this dataset only as auxiliary due to limited positional variety of the target speech

– most of the speakers have been speaking from the front side with respect to the microphones. As such, the dataset is less ideal for exploration of spatial characteristics and signal alignment purposes.

### 4.1.3 MODEL SETTINGS AND EVALUATION METRICS

In this paper, we set the number of channels of the AFnet (Figure 1) as: $C_1, C_2, C_3, C_4 = 32, 64, 64, 64$, resulting in a model size of approximately 2.6M parameters. The slope of leaky ReLU is set to $0.01$. Regarding the STFT processing of the input signal, for the primary dataset we use the Hann window with 512-point fast Fourier transform (FFT) and for CHiME-3 with 1024-point FFT, both using a hop size of 256. During training, 4-second long segments are randomly cropped from the training data samples while during testing the whole utterances are used. The RTFs are obtained by computing the ratios between the clean speech signals at all microphones according to (4). We use the following three commonly seen SE metrics for evaluation purposes. **PESQ:** Perceptual Evaluation of Speech Quality (ITU-T Recommendation P.862.2, 2005). **STOI:** Short-Time Objective Intelligibility (Taal et al., 2011). **SSNR:** Segmental Signal-to-Noise Ratio (Hansen & Pellom, 1998) (segment length = 30 msec, 75% overlap, $\text{SNR}_{\min} = -10$ dB, $\text{SNR}_{\max} = 35$ dB). For all the metrics, the higher the score, the better the performance.

More details of experiential settings can be found int Appendix B.

### 4.2 RESULTS

### 4.2.1 ALIGNED VS. UNALIGNED SIGNAL ENHANCEMENT

To get an understanding of the importance of alignment, we first conduct an experiment to compare the difference between enhancement outcomes of aligned and unaligned input signals with the primary dataset. To this end, two schemes are considered, where in the first scheme we manually align the noisy microphone signals by multiplying them with the corresponding ground truth speech RTFs, while in the other scheme we directly use the original unaligned signals. In each scheme, a single U-Net model, or equivalently, the Filter Net (the second half of AFnet), is trained for the signal reconstruction loss $\mathcal{L}_{\text{rec}}$ and tested on the corresponding aligned or unaligned data. The performance numbers of the two cases are presented in Table 1. It can be seen that when the signals are aligned, the Filter Net model can do a much better job of separating the target signal from background noise and achieves significantly better scores than unaligned results in all three metrics. This observation is consistent across all three number of microphone settings. The results suggest that for a spatially diverse dataset where the target speech may come from various locations, having a signal alignment module in front of the signal filtering module can be beneficial.

We also present the results for the CHiME-3 dataset in Table 1. We can see that the margin between the results of using the aligned and unalinged signals is much smaller here. It could be attributed to that most of the CHiME-3 recording data samples have the speaker almost speaking from the front side with respect to the microphones which results in less variety in directional angles (the common situation when the speaker was asked for holding a tablet to speak the sentence listed in the tablet, and the speech data were recorded through 6 embedded microphones in the same tablet. See the recording setup and recording demonstration picture in CHiME-3 official website) and thus the less spatial diversity. Moreover, as the speech sources mostly come from the front side, they are in a sense already roughly aligned. Thus, there is the less benefit brought by performing the alignment, meaning that the requirement of aligning signals is bypassed to some extend.

Table 1: SE performance comparison between aligned (A) and unaligned (U) signals.

| # Mic | PESQ | | STOI | | SSNR | |
|---|---|---|---|---|---|---|
| | A | U | A | U | A | U |
| 2 | 2.68 | 1.68 | 0.842 | 0.697 | 7.86 | 4.66 |
| 4 | 2.89 | 1.67 | 0.862 | 0.696 | 8.28 | 4.56 |
| 8 | 3.04 | 1.76 | 0.879 | 0.719 | 8.66 | 4.68 |
| CHiME-3 | 2.62 | 2.42 | 0.973 | 0.963 | 10.32 | 9.13 |

### 4.2.2 EXPLOITING SPATIAL SEPARABILITY WITH AFNET

Having the above insight, we move on to studying the proposed AFnet for exploitation of spatial separability on the primary dataset with different model/training settings and comparing with typical direct filter weight estimation approaches. For detailed analysis, we consider four cases below:

i) **U-Net for filter weights:** using only a single U-Net (same model as the Filter Net of AFnet) to directly estimate the sets of filter weights $\mathbf{W}_i$ in typical multichannel SE

ii) **W-Net for filter weights:** using a W-Net (same model as the AFnet, but the output of the first U-Net block goes directly to the next U-Net block without multiplying the input signals) to directly estimate the sets of filter weights $\mathbf{W}_i$ in typical multichannel SE

iii) **AFnet w/o RTF loss:** the two-stage sequential masking AFnet model trained by normal training scheme without incorporating RTFs

iv) **AFnet:** the two-stage sequential masking AFnet model trained with RTF-aware scheme

Cases i) and ii) represent the typical approach which uses a DNN to directly estimate the beamformer weights. Cases iii) and iv) are considered to observe the role of RTF alignment in the AFnet.

Figure 2 presents the enhancement results. One can see that the typical approaches of U-Net and W-Net for direct filter weights estimation do not seem to actually benefit from the multichannel input data, as the performance is not necessarily improving with increased number of microphones. In fact, the W-Net even performs worse than the single U-Net model for the cases with more microphones (4-mic, 8-mic) though doubling the model size, with its best result surprisingly reached for the 2-mic case. This observation implies that to further take advantage of audio data with more microphone channels, simply stacking U-Net models may not straightforwardly work well. This might be due to the potential optimization difficulty of having a deeper model and more complicated multichannel input data to extract useful information. On the other hand, the AFnet seems to mitigate these issues by introducing the sequential masking mechanism which also provides certain input signal level skip connections, as we can see that even without RTF supervision it is able to perform much better than the W-Net. Nevertheless, the performance of AFnet without RTF supervision does not necessarily improve with increased number of microphones. This may be a result of lacking guidance on learning meaningful spatial features. Finally we see that with the RTF loss incorporated for explicitly leveraging spatial separability, the AFnet is able to obtain better performance as the number of microphones increases, and achieves the best performance overall.

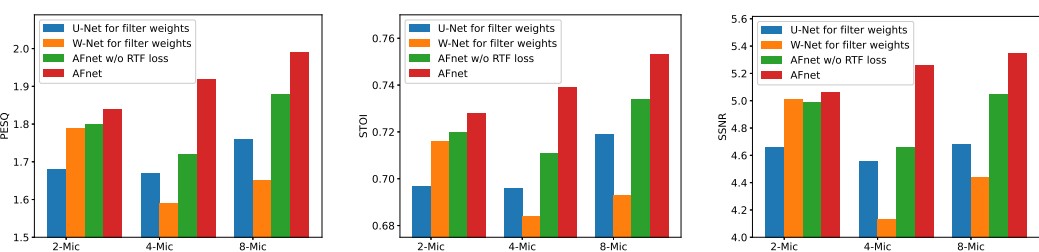

Figure 2: SE performance of different networks and training settings.

### 4.2.3 VISUALIZING THE ALIGNMENT OUTCOMES

To further observe the effect of RTF supervision, we visualize the distributions of the alignment masks in Figure 3 from AFnet trained with RTF loss and that without RTF loss. To obtain these plots, we take noisy signal samples from the test set in which the speech source in each sample is located at one of three pre-defined positions in the room. We feed the taken noisy samples to the trained AFnet models (both with and without RTF supervision) and obtain the alignment masks for each sample. The distribution of the principal components (PCs) of the obtained masks are plotted. It can be seen that with RTF-aware training, the distributions show clearer clusters of speech sources corresponding to the three different positions than with normal training. In addition, with increased number of microphones the clusters also become more separate. The results together with improved enhanced speech quality provide evidence of the fact that the system is actually exploiting spatial separability of the audio sources for achieving better enhancement performance.

### 4.2.4 COMPARISON TO SOTA METHODS

To demonstrate the superiority of our method, we compare the AFnet with several SOTA architectures, including: **Conv-TasNet** (Luo & Mesgarani, 2019): a single-channel SE system performing masking-based separation method in the learned transform domain via learnable encoder and decoder. **DCUnet** (Choi et al., 2018): a phase-aware single-channel SE approach utilizing complex-valued operations with the U-Net architecture. **FaSNet** (Luo et al., 2020): a time-domain, multi-channel SE system exploiting cross-channel correlation based on filter-and-sum method.

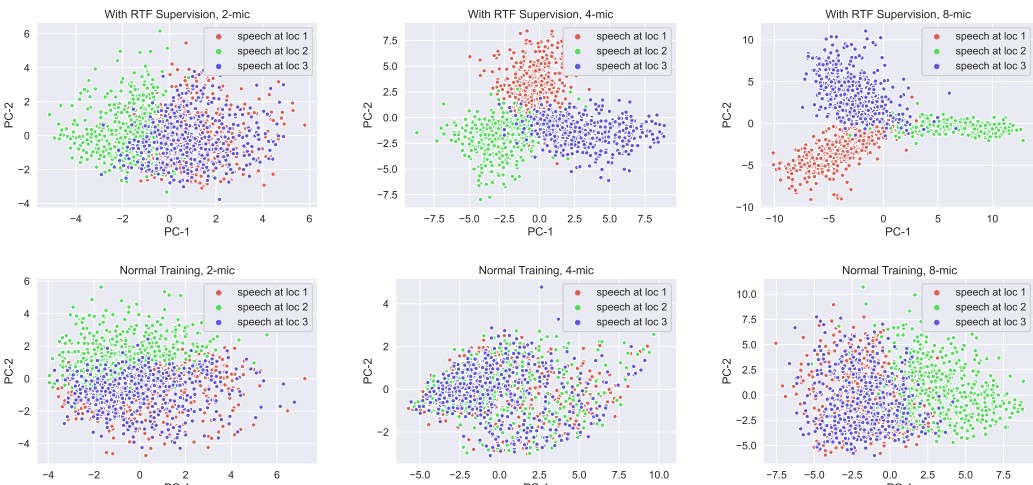

Figure 3: Visualization of the distributions of alignment masks. The RTF-aware training cases (Upper Row) show obvious clusters of signals coming from the three different locations (loc 1, loc 2, loc 3) as the number of microphone increases, while the normal training cases (Lower Row) do not present clear boundaries.

We train the SOTA models by ourselves. For fairness, all models are trained by minimizing the reconstruction loss $\mathcal{L}_{\text{rec}}$ of the combined power-law compressed MSE of (7), using 512-point FFT with a hop size of 256 via the Hann window. The results are presented in Table 2. It can be seen that the single-channel methods do not perform well even with a larger model size. The FaSNet, with a lighter model, performs slightly better than the two single-channel approaches when the number of microphones is larger. Finally, our proposed AFnet outperforms all the SOTA approaches by significant amount in all the metrics used, while being of the smallest model size. Note that the performance of AFnet lies in between the aligned and unaligned signal cases in Table 1, which is reasonable as the alignment is never perfect but only to certain degree.

Although a less ideal dataset for our study on spatial characteristics, we still benchmark the AFnet with several SOTAs on the CHiME-3 dataset, including: **Neural BF** (Erdogan et al., 2016), **MVDR**$_{GC}$ (Bu et al., 2019), **rSDFCN** (Liu et al., 2020), **CA Dense U-Net** (Tolooshams et al., 2020), and **IC Conv-TasNet** (Lee et al., 2021). In Table 3 we report the PESQ and STOI taken from the corresponding papers and the missing entries in the table indicate that the metric is not reported in the reference paper (we do not report SSNR as most papers do not report it). We present both results of the AFnet trained with and without RTF supervision (with the 5-th channel selected as the reference microphone). One can see that the two-stage, sequential masking design of the AFnet is efficient in that it achieves comparable or better performance to other SOTA approaches while maintaining a small model size. Notably, the RTF supervision helps improve the AFnet performance, while the gains are minor in this case due to the less spatial diversity of the speech data. Nevertheless, the results still show the benefit of incorporating signal alignment into the network design, even though on a dataset with limited positional variety of the speech target.

### 4.2.5 IMPORTANCE OF PHASE ALIGNMENT IN RTF ESTIMATION

The Align Net trained with the RTF loss performs both temporal alignment as well as level adjustment. It would be interesting to look at how the magnitude and phase components of the RTFs, which correspond to ILD and ITD in time domain, affect the enhancement performance of the AFnet. In Figure 4 we compare four different RTF alignment schemes – no alignment, aligning RTF magnitude only, aligning RTF phase only, and aligning everything for the RTF loss $\mathcal{L}_{\text{rtf}}$. From the results it can be seen that phase alignment is the key to obtaining improved performance over the alignment-agnostic system. This is reasonable as aligning phase in the frequency domain corresponds to compensating the delay in the time domain, which is crucial for any beamforming algorithms that rely on spatial separability. The results suggest that the AFnet indeed exploits spatial information for improved enhancement performance.

Table 2: SE comparison with SOTA methods on the primary dataset.

| Methods | # Params | PESQ | STOI | SSNR |
|---|---|---|---|---|
| Noisy | - | 1.21 | 0.577 | -1.70 |
| Conv-TasNet | 8.7M | 1.47 | 0.636 | 3.00 |
| DCUnet | 7.6M | 1.49 | 0.660 | 2.91 |
| FaSNet (2-mic) | | 1.55 | 0.663 | 3.70 |
| FaSNet (4-mic) | 2.8M | 1.57 | 0.667 | 3.64 |
| FaSNet (8-mic) | | 1.64 | 0.683 | 3.66 |
| AFnet (2-mic) | | 1.84 | 0.728 | 5.06 |
| AFnet (4-mic) | 2.6M | 1.92 | 0.739 | 5.26 |
| AFnet (8-mic) | | 1.99 | 0.753 | 5.35 |

Table 3: SE comparison with SOTA methods on the CHiME-3 dataset.

| Methods | # Params | PESQ | STOI |
|---|---|---|---|
| Noisy | - | 1.27 | 0.870 |
| Neural BF | - | 2.29 | - |
| $MVDR_{GC}$ | 0.5M | - | 0.952 |
| rSDFCN | 2.1M | 2.15 | 0.937 |
| CA Dense U-Net | >20M | 2.44 | - |
| IC Conv-TasNet | 1.7M | 2.67 | 0.973 |
| AFnet w/o RTF Loss (ours) | 2.6M | 2.68 | 0.972 |
| AFnet (ours) | | 2.72 | 0.972 |

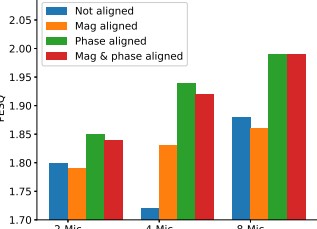 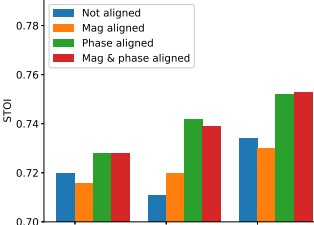 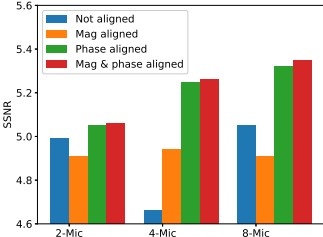

Figure 4: SE comparison of different RTF alignment schemes for AFnet training.

### 4.2.6 ARE INTERMEDIATE ALIGNMENT MASKS NECESSARY?

Given the results in Figure 4 where the phase importance is recognized, one would ask if the intermediate alignment mask design of AFnet is necessary, and if we can directly apply phase alignment at the output filtering masks? To answer this question, we train the W-Net for predicting the filter weights (i.e., the filtering masks) to minimize $\mathcal{L}_{rec} + \lambda\mathcal{L}_{rtf, phase}$, where $\mathcal{L}_{rec}$ is the same signal reconstruction loss as (7) and $\mathcal{L}_{rtf, phase}$ is the RTF alignment loss similar to (6) but imposed on the filtering masks given by the W-Net for the phase portion only and regularized by $\lambda > 0$. Several values of $\lambda$ are considered and the results are compared to AFnet in Table 4. We can see that over a wide range of $\lambda$ the W-Net is still not able to achieve as good performance as AFnet which utilizes the alignment masks with RTF supervision. This implies that the intermediate masking mechanism for performing alignment is critical for achieving improved SE with two-stage network design.

Table 4: SE comparison of AFnet and W-Net with RTF phase regularization on the filtering masks.

| Method | PESQ | | | STOI | | | SSNR | | |
|---|---|---|---|---|---|---|---|---|---|
| | 2-mic | 4-mic | 8-mic | 2-mic | 4-mic | 8-mic | 2-mic | 4-mic | 8-mic |
| W-Net for filter weights, no phase reg. | 1.74 | 1.61 | 1.67 | 0.703 | 0.679 | 0.688 | 4.80 | 4.07 | 4.39 |
| W-Net for filter weights, phase reg. $\lambda = 0.0001$ | 1.76 | 1.61 | 1.65 | 0.713 | 0.686 | 0.692 | 4.95 | 4.21 | 4.42 |
| W-Net for filter weights, phase reg. $\lambda = 0.001$ | 1.63 | 1.60 | 1.68 | 0.688 | 0.686 | 0.702 | 4.36 | 4.14 | 4.60 |
| W-Net for filter weights, phase reg. $\lambda = 0.01$ | 1.71 | 1.59 | 1.61 | 0.705 | 0.683 | 0.684 | 4.55 | 4.14 | 4.30 |
| W-Net for filter weights, phase reg. $\lambda = 0.1$ | 1.73 | 1.66 | 1.74 | 0.708 | 0.701 | 0.710 | 4.09 | 3.54 | 3.87 |
| AFnet | 1.84 | 1.92 | 1.99 | 0.728 | 0.739 | 0.753 | 5.06 | 5.26 | 5.35 |

Additional results on a variety of testing scenarios, e.g., SNR conditions, room configurations, training protocols, time-varying RIR and realistic RIR data are provided in Appendix C. Comparison of Align Net with signal processing-based algorithms for RTF estimation is presented in Appendix D.

## 5 CONCLUSION

In this paper, we presented the AFnet, a deep learning-based multichannel SE approach that performs signal alignment followed by filtering for extracting the target speech from noisy observations. We showed that leveraging spatial information inherent in the RTFs for signal alignment purposes of the two-stage, sequential masking network during training is the key to remarkable improvements. Our findings suggest that RTF alignment, especially the phase estimation, plays a crucial role in deep learning multichannel SE for target speech that may come from arbitrary locations.

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

## APPENDIX A   CONVENTIONAL SIGNAL PROCESSING BEAMFORMING ALGORITHMS BASED ON ALIGN-THEN-FILTER PROCESS

To support the motivation behind the proposed AFnet which is designed based on the align-then-filter concept well-known in signal processing, we review several spatial filtering algorithms based on using this design principle.

The intuition for performing alignment followed by filtering is relatively simple. Consider the RTF aligned signal expression in (5). We can see that in the aligned signals each $Z_i(f, t)$ contains $S_r(f, t)$. Since all the $S_r(f, t)$ among the $N$ channels are in-phase (and actually are identical if magnitudes are also aligned in this case), summing up all the $Z_i(f, t)$ over the $N$ channels will not be destructive to the speech component. However, the noise components $\tilde{V}_i(f, t)$ are usually not in-phase (unless coming from the same direction as the speech source), destructive combination can happen by summing them up. Consequently, combining all the $Z_i(f, t)$ across the $N$ channels will boost the SNR as the speech component is preserved while the noise components are suppressed.

The above process is actually the classic "delay-and-sum (DS)" beamformer Trees (2004); Doclo et al. (2015). In a DS beamformer, a steering vector containing the time delay information between the microphone sensors with respect to the target speech is first applied to the input multichannel signals to compensate the time delay (or equivalently the phase difference) of the speech components in all channels. This step is called "alignment." Then a summation operation over the aligned channels is performed to suppress noise for boosting SNR. This step is called "filtering." Mathematically, the enhancement process of the DS beamformer can be expressed as:

$$\hat{S}(f, t) = \mathbf{h}^T(f, t)\mathbf{x}(f, t), \tag{8}$$

$\forall f, t$, where $\tilde{\mathbf{h}}(f, t) = [\tilde{H}_1(f, t), \dots, \tilde{H}_N(f, t)]^T$ is the steering vector consisting of RTFs (potentially only the phase components) and $\mathbf{x}(f, t) = [X_1(f, t), \dots, X_N(f, t)]^T$ is the vector of microphone signals.

Another famous beamformer utilizing the align-then-filter concept is the "generalized sidelobe canceller (GSC)" (Griffiths & Jim, 1982; Frost, 1972). In a typical GSC, the input noisy signals are first aligned by the steering vector. Then, two branches of processing follow, where one branch sums up all the aligned microphone signals to perform a DS beamforming, while the other branch introduces a "blocking matrix" to extract noise references while blocking out the target speech component. Then, the noise-only output of the blocking matrix will go through an adaptive filtering algorithm to estimate the residual noise component in the DS beamforming output signal in an adaptive manner. Note that the alignment step plays an important role here. First, it leads to the DS beamformer for boosting SNR. In addition, it is beneficial for the blocking matrix stage to extract noise-only references. For example, a simple blocking mechanism can be just subtracting all other $Z_i(f, t)$, $i \neq r$ from the reference channel $Z_r(f, t)$ so that the $S_r(f, t)$ will be cancelled out and only the noise remains. The GSC is obviously another "align-then-filter" example with a more sophisticated filtering stage.

Finally, the popular "minimum-variance-distortionless-response (MVDR)" beamformer (Capon, 1969; Doclo et al., 2015) can also belong to the align-then-filter framework. The derivation of the MVDR filter weights $\mathbf{w} = [W_1(f, t), \dots, W_N(f, t)]^T \in \mathbb{C}^N$ which applies to the noisy signals to obtain the enhanced speech as $\hat{S}(f, t) = \mathbf{w}^H(f, t)\mathbf{x}(f, t)$ starts from considering the following optimization problem:

$$\min_{\mathbf{w}} \mathbf{w}^H \mathbf{\Phi}_v(f, t)\mathbf{w}, \quad \text{subject to } \mathbf{w}^H \bar{\mathbf{h}}(f, t) = 1, \tag{9}$$

where $\mathbf{\Phi}_v(f, t) = E[\mathbf{v}(f, t)\mathbf{v}(f, t)^H] \in \mathbb{C}^{N \times N}$ is the power spectral density of the vector of noise signals $\mathbf{v}(f, t) = [V_1(f, t), \dots, V_N(f, t)]^T$ and $\bar{\mathbf{h}}(f, t) = [\tilde{H}_1^{-1}(f, t), \dots, \tilde{H}_N^{-1}(f, t)]^T$ is the (reciprocal) vector of RTFs. Solving the above we obtain the MVDR filter as:

$$\mathbf{w}(f, t) = \frac{\mathbf{\Phi}_v^{-1}(f, t)\bar{\mathbf{h}}(f, t)}{\bar{\mathbf{h}}^H(f, t)\mathbf{\Phi}_v^{-1}(f, t)\bar{\mathbf{h}}(f, t)}, \tag{10}$$

$\forall f, t$. We see that the RTFs which are related to signal alignment are involved in the computation of the MVDR filter weights.

From the above, we can see that the RTF information plays an essential role in many conventional beamforming algorithms. This has motivated us to devise the AFnet following the alignment principle. Note that the signal processing-based beamforming algorithms typically rely on modeling assumptions of signal or noise statistics to perform denoising, and therefore the performance is limited due to mismatch of real-world observations and modeling assumptions. Our approach is based on data-driven, deep learning techniques to overcome such limitations while taking advantage of the RTFs.

## APPENDIX B    IMPLEMENTATION DETAILS

### B.1    RTF-AWARE TRAINING ALGORITHMS

The main RTF-incorporated training scheme is summarized in Algorithm 1, which performs two-step training consisting of i) training the Align Net for the RTF loss followed by ii) training both Align and Filter Nets jointly for the signal reconstruction loss.

---

**Algorithm 1** RTF-Aware Training (Two-Step)

---

**inputs**
    Model($\mathbf{X}_i \in \mathbb{C}^{F \times T}, \forall i = 1, \ldots, N$; Align Net, Filter Net)
Initialize Align Net, Filter Net
**for** number of training iterations **do**
    Sample a minibatch of new input spectrograms $\mathbf{X}_i, \forall i = 1 \ldots, N$
    Update Align Net by minimizing $\mathcal{L}_{\text{rtf}}$
**end for**
**for** number of training iterations **do**
    Sample a minibatch of new input spectrograms $\mathbf{X}_i, \forall i = 1 \ldots, N$
    Update Align Net and Filter Net by minimizing $\mathcal{L}_{\text{rec}}$
**end for**

---

Another way of RTF-aware training for the AFnet is by combining the RTF loss and signal reconstruction loss together and jointly train the entire AFnet from scratch. In this sense, the RTF loss serves as a regularization term weighted by a scaler $\lambda > 0$. The training scheme is depicted in Algorithm 2. Later in Section C.1 we will show that this is less effective than Algorithm 1.

---

**Algorithm 2** RTF-Aware Training (via Regularization)

---

**inputs**
    Model($\mathbf{X}_i \in \mathbb{C}^{F \times T}, \forall i = 1, \ldots, N$; Align Net, Filter Net)
Initialize Align Net, Filter Net
**for** number of training iterations **do**
    Sample a minibatch of new input spectrograms $\mathbf{X}_i, \forall i = 1 \ldots, N$
    Update Align Net and Filter Net by minimizing $\mathcal{L}_{\text{rec}} + \lambda \mathcal{L}_{\text{rtf}}$
**end for**

---

### B.2    PRIMARY DATASET

**Corpus:** We use Audio-Visual Speech Dataset (AVSpeech) (Ephrat et al., 2018) available online. AVSpeech is a large collection of video clips of single speakers talking with no audio background interference. The audio files of the AVSpeech dataset serve as proper target speech files due to their variety (diversity of language, gender, age, etc.) The dataset can be found at: `https://looking-to-listen.github.io/avspeech/`.

**Noise data:** We collected 8 types of noise profiles from YouTube which are commonly seen sound sources in daily life. The noise types include blender, vacuum, washer, baby cry (for training) and dog barking, kids playing sound, hair dryer, food sizzling (for testing). Each type of noise profile contains recordings from different sound sources of the same type.

**Data pre-processing:** All the raw audio data are first converted to 16 kHz mono .wav files before feeding to the room acoustic simulator to generate the multichannel data samples for experimentation (the AVSpeech audio originally may contains 1 or 2 channels of audio). For spectral processing, we use the Hann window with 512-point FFT and a hop size of 256 for the STFT. By conjugate symmetry of the FFT, only half of the frequency bins (i.e., 257) are actually needed for the network to process the spectrograms.

**Room acoustic simulator:** Pyroomacoustics is utilized to generate multichannel audio which can be found at `https://github.com/LCAV/pyroomacoustics/tree/pypi-release`. We create a room of size $8 \times 8 \times 3$ (length×width×height in meters). Reverberation time is randomly chosen from {0.16, 0.32, 0.48, 0.64} second. The RIRs are estimated based on the image source method. 8 microphones are placed to form a planar array (4 columns, 2 rows) vertically, where the distance between adjacent microphones is set to 5 cm. The microphone array center is placed at the center of the room. Speech and noise signals are randomly positioned at a distance of {1, 2, 3, 4} meters from the microphone array center. In the experiments, the upper leftmost and rightmost microphones are used for the 2-mic scheme, the microphones at the four corners are used for the 4-mic scheme, and all microphones are used for the 8-mic scheme.

**Real-world measured RIRs:** To further validate the proposed approach's generalization capabilities to more realistic scenarios, we also conduct experiments on multichannel audio data generated using real-world measured acoustic RIRs. A popular multichannel impulse response dataset is from the work by Hadad et al. (2014), where the impulse responses of several 8-channel microphone arrays with respect to various target source directions (from -90 to 90 degrees) are provided. We utilize the 3-3-3-8-3-3-3 array impulse responses of the dataset including reverberation time of 0.16, 0.36, and 0.61 second cases (`https://www.iks.rwth-aachen.de/en/research/tools-downloads/databases/multi-channel-impulse-response-database/`), together with the AVSpeech utterances and the YouTube noise profiles to generate the multichannel noisy data for experiments. The experimental results and related discussion are presented in Appendix C.7.

**Mixing speech with noise:** We mix the multichannel speech files with noise profiles to generate noisy-clean data pairs. Each type of noise is multiplied with a scale randomly chosen from {0, 0.5, 1, 1.5, 2} before added up, and the combined noise is then added to the clean speech according to a specific SNR level randomly selected from {-10, -6, -3, 0, 3, 6, 10} dB. In this way, we are adopting the supervised learning SE scheme of mix-and-separate, where the clean and noise signals are separately collected and subsequently mixed together to become the noisy signals for training purposes. This mix-and-separate scheme by assuming additive noise model is still being used in the majority of SE works (not only signal processing- but also deep learning-based approaches). To apply it to real data, one may collect the clean and noise recordings using the target device in real-world environments and mix them up to generate noisy mixtures for training.

**Network architectures:**

- **AFnet:** the implementation of AFnet (Figure 1) is based on the deep complex U-Net model at: `https://github.com/sweetcocoa/DeepComplexUNetPyTorch`, for both the Align Net and Filter Net parts. The number of layers and the number of features in each layer are modified to the one shown in Figure 1, where $C_1, C_2, C_3, C_4 = 32, 64, 64, 64$.

- **Conv-TasNet**: the experimental results of Conv-TasNet is based on the implementation from `https://github.com/kaituoxu/Conv-TasNet` using their own model settings instead of the original Conv-TasNet paper. Note that although the model hyperparameters were designed based on 8 kHz sampling rate, we directly use them for 16 kHz sampling rate of our data without further modifications. Hyperparameters: $N = 256, L = 20, B = 256, H = 512, P = 3, X = 8, R = 4$, Norm=gLN, Noncausal.

- **DCUnet**: the experimental results of DCUnet is based on the implementation of the code at `https://github.com/mhlevgen/DCUNetTorchSound`. We choose the Large-DCUnet-20 model which realizes the 20-layer model in the original DCUnet paper. Note that although in the original system a 1024-point FFT was used, here we use 512-point FFT instead.

- **FaSNet**: the experimental results of FaSNet is based on the implementation of the code at: `https://github.com/yluo42/TAC`. We choose to use the FaSNet_TAC model where TAC stands for transform-average-concatenate. Hyperparameters: enc_dim=64, feature_dim=64, hidden_dim=128, layer=4, segment_size=50, nspk=1, win_len=4, context_len=16, sr=16000.

**Model optimization:** All experiments were run on one NVIDIA Tesla V100 GPU of 32 GB CUDA memory.

- **AFnet training:** the AFnet is trained via the RTF-aware training scheme. The Adam optimizer is used for minimizing $\mathcal{L}_{\text{rtf}}$ with a total of 80 epochs, where for the first 50 epochs a learning rate of 0.001 is used and for the rest 30 epochs it is decreased to 0.0001. For minimizing $\mathcal{L}_{\text{rec}}$ we train the network for another 55 epochs with Adam, where for the first 50 epochs a learning rate of 0.001 is used and for the rest 5 epochs it is decreased to 0.0001. The total number of epochs is 80+55=135. A batch size of 4 is used.

- **AFnet training (w/o RTF supervision):** the AFnet trained without RTF loss is just minimizing $\mathcal{L}_{\text{rec}}$. In this case, the Adam optimizer with a learning rate of 0.001 is used, which is decreased to 0.0001 at the 50-th epoch, and with a total of 80 epochs. A batch size of 4 is used.

- **Conv-TasNet, DCUnet, and FaSNet training:** for fair comparison purposes, the training of the SOTA methods is same by minimizing the reconstruction loss $\mathcal{L}_{\text{rec}}$ using 512-point FFT with a hop size of 256 via the Hann window (same STFT setting as AFnet). For the time-domain approaches (Conv-TasNet and FaSnet), the waveform of the model output is converted to STFTs for computing the loss. In all cases, the Adam optimizer with a learning rate of 0.001 is used, which is decreased to 0.0001 at the 50-th epoch, and with a total of 80 epochs. A batch size of 4 is used.

### B.3 CHiME-3 DATASET

**Dataset:** The publicly available CHiME-3 dataset (Barker et al., 2015), made available as part of a speech separation and recognition challenge, is used for training and evaluating SE performance. The dataset is a 6-channel microphone recording of talkers speaking in a noisy environment, sampled at 16 kHz. It consists of 7,138 and 1,320 simulated utterances with an average length of 3 seconds for training and testing, respectively.

**SOTAs to compare:**

- **Neural BF** (Erdogan et al., 2016): An MVDR beamforming with mask estimation through bidirectional-LSTM.

- **MVDR**$_{GC}$ (Bu et al., 2019): An MVDR beamforming using a neural network-based method to identify and correct phase errors in the steering vector.

- **rSDFCN** (Liu et al., 2020): A time-domain, fully convolutional network (FCN) with sinc and dilated convolutional layers for multichannel SE.

- **CA Dense U-Net** (Tolooshams et al., 2020): A time-frequnecy domain multichannel SE model that combines the merits of DenseNet, U-Net, and channel attention (CA) mechanism.

- **IC Conv-TasNet** (Lee et al., 2021): The Inter-Channel (IC) Conv-TasNet is the extension of the time-domain Conv-TasNet for single-channel SE to the multichannel SE case.

**Spectral processing:** We use the Hann window with 1024-point FFT and a hop size of 256 for the STFT. By conjugate symmetry of the FFT, only half of the frequency bins (i.e., 513) are actually needed for the network to process the spectrograms.

**Model optimization:** The AFnet training was run on one NVIDIA Tesla V100 GPU of 32 GB CUDA memory using the same model optimization scheme discussed in Section B.2.

### B.4 Evaluation metrics

**PESQ:** a speech quality measure using the wide-band version recommended in ITU-T P.862.2. It basically models the mean opinion scores (MOS) that cover a scale from 1 (bad) to 5 (excellent). We use the Python-based PESQ implementation from: `https://github.com/ludlows/python-pesq`.

**STOI:** a function that well represents the average intelligibility of the degraded speech. It provides a value from 0 to 1, which can be interpreted as the percentage of correctly recognized words by normal-hearing people. We use the Python-based STOI implementation from: `https://github.com/mpariente/pystoi`.

**SSNR:** an SNR measure, instead of working on the whole signal, that calculates the average of the SNR values of short segments (segment length $= 30$ msec, $75\%$ overlap, $\text{SNR}_{\min} = -10$ dB, $\text{SNR}_{\max} = 35$ dB). We use the Python-based SSNR implementation from: `https://github.com/schmiph2/pysepm`.

## Appendix C  Additional experimental results

### C.1 Comparison of RTF-aware training schemes (two-step vs. regularized)

In this section we try to answer whether the two-step RTF-aware training scheme (Algorithm 1) or the regularized training scheme (Algorithm 2) is better. To this end, we train the AFnet model with several values of $\lambda$ using Algorithm 2 and the results are compared to AFnet trained with Algorithm 1 in Table 5. It can be seen that both training schemes help improve over the RTF-agnostic results (i.e., $\lambda = 0$) as they both incorporate RTF information for explicitly learning spatial characteristics. However, the two-step approach performs consistently better than the regularized approach. The results suggest that learning to align prior to learning to denoise during the training stage is more beneficial.

Table 5: SE performance comparison of RTF-aware training schemes for AFnet.

| Method | PESQ | | | STOI | | | SSNR | | |
|---|---|---|---|---|---|---|---|---|---|
| | 2-mic | 4-mic | 8-mic | 2-mic | 4-mic | 8-mic | 2-mic | 4-mic | 8-mic |
| Algorithm 1 | 1.84 | 1.92 | 1.99 | 0.728 | 0.739 | 0.753 | 5.06 | 5.26 | 5.35 |
| Algorithm 2, $\lambda = 0$ | 1.80 | 1.72 | 1.88 | 0.720 | 0.711 | 0.734 | 4.99 | 4.66 | 5.05 |
| Algorithm 2, $\lambda = 0.0001$ | 1.77 | 1.77 | 1.88 | 0.718 | 0.714 | 0.735 | 4.80 | 4.76 | 5.10 |
| Algorithm 2, $\lambda = 0.001$ | 1.80 | 1.73 | 1.88 | 0.721 | 0.714 | 0.735 | 4.98 | 4.64 | 5.10 |
| Algorithm 2, $\lambda = 0.01$ | 1.78 | 1.82 | 1.84 | 0.718 | 0.729 | 0.726 | 4.91 | 4.94 | 4.98 |
| Algorithm 2, $\lambda = 0.1$ | 1.82 | 1.82 | 1.95 | 0.723 | 0.726 | 0.744 | 5.09 | 4.94 | 5.28 |
| Algorithm 2, $\lambda = 1$ | 1.76 | 1.79 | 1.88 | 0.713 | 0.720 | 0.731 | 4.81 | 4.89 | 5.13 |

### C.2 Fixing vs. unfixing Align Net while minimizing reconstruction loss

We compare two additional training schemes of the AFnet – fixing or unfixing the Align Net parameters after it has been trained via the RTF loss $\mathcal{L}_{\text{rtf}}$, when we train the full AFnet for the reconstruction loss $\mathcal{L}_{\text{rec}}$. Table 6 presents the results. It can be seen that adjusting the Align Net parameters together with the Filter Net parameters for optimizing the reconstruction loss is crucial for achieving better performance compared to fixing the Align Net after the RTF loss minimization phase. This indicates that both stages may share the task of performing spatial and spectral denoising jointly, where the first stage shares more loading on the spatial denoising part while the second stage is more on the spectral filtering portion. Dividing the SE task into the two subtasks makes the network learn to more efficiently denoise the speech.

Table 6: SE performance comparison between using fixed (F) and unfixed (U) Align Net for training the AFnet after the Align Net has been trained with the RTF loss.

| # Mic | PESQ | | STOI | | SSNR | |
|---|---|---|---|---|---|---|
| | F | U | F | U | F | U |
| 2 | 1.77 | 1.84 | 0.712 | 0.728 | 4.93 | 5.06 |
| 4 | 1.81 | 1.92 | 0.721 | 0.739 | 5.01 | 5.26 |
| 8 | 1.84 | 1.99 | 0.728 | 0.753 | 4.99 | 5.35 |

### C.3 VISUALIZING THE DATASET SPATIAL DIVERSITY

Given the ability of AFnet to exploit spatial separability, it is also interesting to utilize the AFnet for inspecting the spatial diversity of the dataset. Such information can be provided by observing the alignment masks of the data points. In Figure 5 we compare the distributions of the intermediate alignment masks learned by the AFnet models trained on the primary dataset (high spatial diversity) and the CHiME-3 dataset (low spatial variety). Here, we plot the distributions of the alignment masks estimated by the trained models for all the test data samples of the two datasets. It can be seen that for the primary dataset, the RTF-aware training leads to a more widespread distribution than that of the normal training case, corresponding to the greater spatial diversity of the target speech source positions. In contrast, in the CHiME-3 dataset the RTF-aware training actually results in a more aggregated cluster than the normal training scheme, indicating that the target speech sources are coming from similar directions. This visualization of the alignment masks provides another angle to inspect the data properties from the spatial diversity aspect for exploring SE performance.

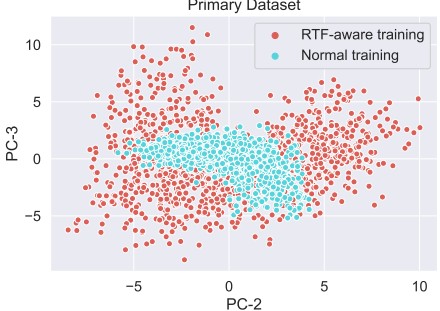 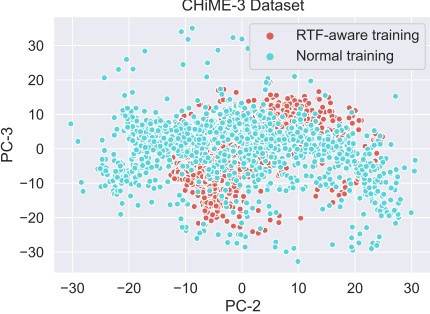

Figure 5: Visualization of the distributions of alignment masks. (Left): the estimated masks for the primary dataset using AVSpeech audio with Pyroomacoustics. (Right): the estimated masks for the CHiME-3 dataset.

### C.4 ENHANCEMENT PERFORMANCE VS. INPUT SNR CONDITION COMPARISON

To look more into the testing results, we present the PESQ, STOI, and SSNR performance numbers vs. input SNR conditions in Figure 6 for the three schemes:

i) **AFnet:** the two-stage AFnet model trained with the RTF-aware training scheme
ii) **AFnet w/o RTF loss:** the two-stage AFnet model trained with normal training scheme without incorporating RTFs
iii) **W-Net for filter weights:** using a W-Net (same model as the AFnet, but the output of the first U-Net block goes directly to the next U-Net block without multiplying the input signals) to directly estimate the sets of filter weights $\mathbf{W}_i$ in typical multichannel SE

We take 160 utterances for each specified input SNR value from the test set and obtain the corresponding performance number of each input SNR value. The results for the 8-mic systems are presented. From the results we can see that the proposed AFnet consistently improves the baseline method over a wide range of input SNR conditions.

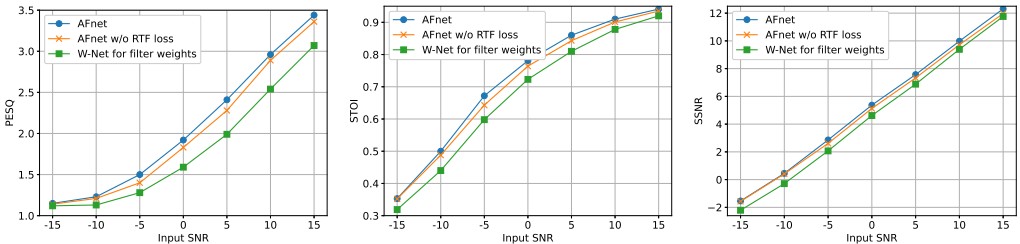

Figure 6: SE performance vs. input SNR conditions.

## C.5 GENERALIZATION TO OTHER ROOM CONFIGURATIONS

To further test the generalization of the proposed AFnet to unseen room configurations, we utilize the Pyroomacoustic simulator to generate test data randomly sampled from three different room sizes: $(8 \times 5 \times 3)$, $(7 \times 6 \times 4)$, and $(8 \times 6 \times 3)$ in terms of (length×width×height) in meters, which are different from the training room (i.e., $(8 \times 8 \times 3)$). Figure 7 presents the SE results under the unseen room configurations. We can see the the proposed AFnet generalizes well to unseen room schemes as it still improves over the typical direct filter weight estimation approaches in most cases. Moreover, the performance of AFnet consistently improves as the number of microphones increases.

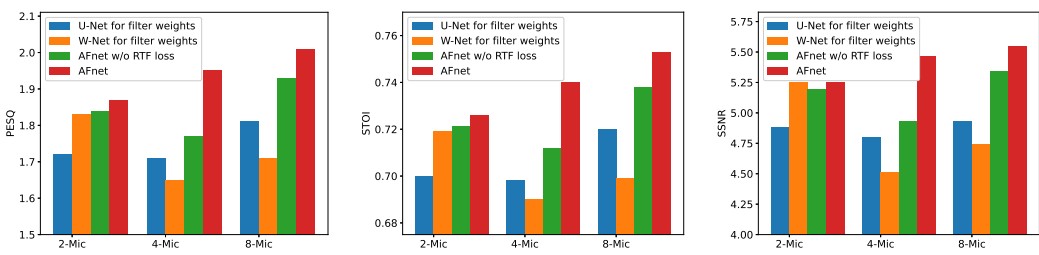

Figure 7: SE performance generalization to unseen room configurations.

## C.6 SE UNDER TIME-VARYING RIR SCENARIOS

The above results have been tested on a relatively static environments where the target speech position of an utterance, once given, is not changing. However, in most real-world scenarios the speaker may naturally move around and therefore the RIRs will not always be time-invariant. To test the proposed AFnet under such circumstances, we leverage the Pyroomacoustic simulator to generate another set of test data (same amount of data as the static RIR case in the previous sections) in which during an utterance the RIRs are changed from one set to another to emulate speaker movements. We evaluate the algorithms on such time-varying RIR generated multichannel dataset and the results are presented in Figure 8. Note that the testing is performed on the same models that have been trained only with the data generated with static RIRs before. From the results we can see that the proposed AFnet has a better tracking ability to tackle time-varying target speech locations, even though the model has not observed any changing RIR data during the training stage. This indicates that the RTF supervision of the alignment masks is also crucial for SE under time-varying RIR environments – by incorporating phase alignment into the model learning, the positional information of the target speech can be better captured for improved SE even under the moving target scenario.

## C.7 RESULTS ON REAL-WORLD MEASURED RIR GENERATED DATA

The results presented in previous sections on the primary dataset are based on the multichannel audio data synthesized by using the image source method via the Pyroomacoustics library, which is useful for methodological study but also has its limitation to represent real data. To further validate the proposed approach's generalization capabilities to more realistic scenarios, we also conduct

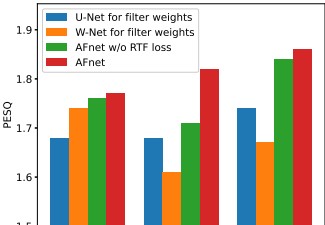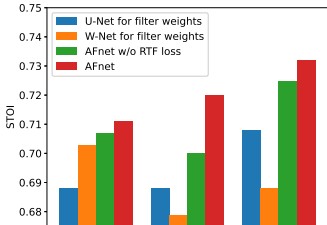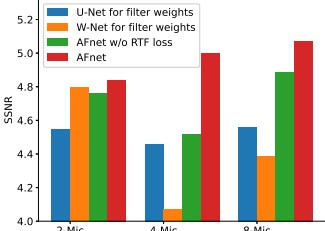

Figure 8: SE performance under time-varying RIR scenarios.

experiments on multichannel audio data generated using real-world measured acoustic RIRs taken from the work of Hadad et al. (2014), where the impulse responses of several 8-channel linear microphone arrays with respect to various target source directions of arrival (DoAs) (from -90 to 90 degrees) are provided. We utilize the 3-3-3-8-3-3-3 array impulse responses of the dataset including reverberation time of 0.16, 0.36, and 0.61 second cases, together with the AVSpeech utterances and the YouTube noise profiles to generate the multichannel noisy data for experiments. To observe the relation of the AFnet performance with the speech source spatial diversity, we generate a spatially diverse dataset where the speech DoA ranges from -90 to 90 degrees, and another spatially constant dataset where the speech comes from only one direction.

Table 7 and Table 8 present the results for the two datasets respectively. One can see that for for the spatially diverse dataset of Table 7, the AFnet clearly benefits from the sequentially masking mechanism and the RTF loss supervision, while for the spatially constant dataset of Table 8 the three models perform comparably with the RTF loss marginally improves AFnet. These results show that a model (such as the W-Net) that performs well on a dataset lacking spatial variety of the target speech may not generalize well to a spatially diverse dataset, unless further attention is paid to exploiting spatial characteristics of the speech sources. The RTF information leveraged by the two-stage sequential masking design of the AFnet is shown to take advantage of the spatial separability for enhanced SE performance on spatially diverse datasets.

Table 7: Results on spatially diverse multichannel data generated using measured acoustic RIRs where the target speech DoA range is $[-90, 90]$ degrees.

| Methods | PESQ | STOI | SSNR |
|---|---|---|---|
| Noisy | 1.40 | 0.598 | -0.91 |
| W-Net for filter weights | 1.89 | 0.693 | 3.20 |
| AFnet w/o RTF Loss | 2.06 | 0.728 | 3.83 |
| AFnet | 2.22 | 0.759 | 4.16 |

Table 8: Results on spatially constant multichannel data generated using measured acoustic RIRs where the target speech DoA is fixed to one direction.

| Methods | PESQ | STOI | SSNR |
|---|---|---|---|
| Noisy | 1.42 | 0.609 | -0.51 |
| W-Net for filter weights | 1.93 | 0.708 | 3.81 |
| AFnet w/o RTF Loss | 1.93 | 0.702 | 3.95 |
| AFnet | 1.99 | 0.717 | 3.40 |

Similar to what have been done in Figure 3 for the simulated RIR case, we also visualize the estimated alignment masks for the spatially diverse dataset of the realistic RIR case. The results in Figure 9 once again show that the RTF-aware training results in clear clusters of the speech sources coming from difference directions as compared to the normal training case, confirming that the network is more aware of the spatial separability of the sound sources.

## APPENDIX D   ALIGN NET VS. CONVENTIONAL SIGNAL PROCESSING APPROACHES FOR RTF ESTIMATION

In this section, we study the effectiveness of the deep leaning-based Align Net for estimating the RTFs compared to conventional model-based signal processing methods in the presence of noise. We show that the Align Net performs reasonably well in the low SNR regime where signal processing approaches fail to, justifying its usage of alignment purposes for later enhancement processing in AFnet.

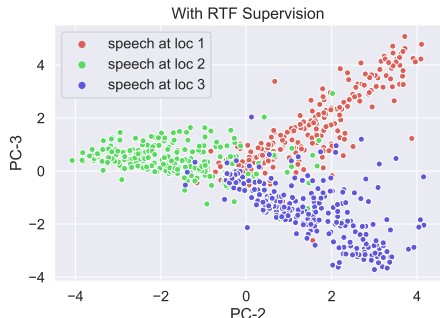 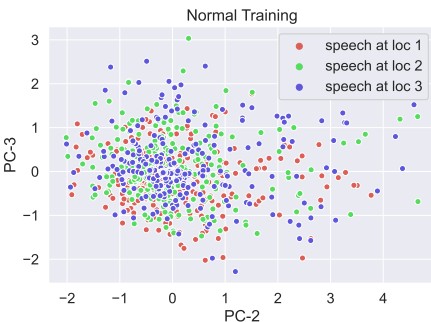

Figure 9: Visualization of the distributions of alignment masks for the spatially diverse data generated using real-world measured acoustic RIRs. Again, the RTF-aware training case (Left) shows obvious clusters of signals coming from the three different locations (loc 1, loc 2, loc 3), while the normal training case (Right) does not present clear boundaries.

### D.1    RTF ESTIMATION PROBLEM

RTFs or their time-domain counterparts, relative impulse responses (ReIRs) are important tools in several multichannel audio processing tasks (Gannot & Cohen, 2004; Laufer et al., 2013). The estimation of RTFs has been studied for a while in the audio signal processing field (Gannot et al., 2001; Koldovskỳ et al., 2015; Giri et al., 2016; 2018; Srikrishnan et al., 2018), where the task is to estimate the correlation between the target source components received by the two microphones, either in terms of RTFs or ReIRs, given the noisy recordings. Following the previous works, we will discuss the scenario with two microphones (i.e., left and right microphones) while the idea may be generalized to more microphones.

Consider a two-channel noisy recording of a target speech in a noisy environment, whose position is fixed for a certain time interval. This situation can be represented as:

$$
\begin{aligned}
x_L(n) &= \underbrace{h_L(n) * s(n)}_{s_L(n)} + v_L(n), \\
x_R(n) &= \underbrace{h_R(n) * s(n)}_{s_R(n)} + v_R(n),
\end{aligned}
\tag{11}
$$

where $n$ is the time sample index taking values $1, \ldots, N$; $*$ denotes the convolution; $x_L$ and $x_R$ are, respectively, the signals from the left ($L$) and right ($R$) microphones; $h_L$ and $h_R$ are the impulse responses between the target and the two microphones; $s$ is the far-field target speech; and $v_L$ and $v_R$ are the noise components.

Let $h_{rel}$ represent the ReIR between the speech signal components arriving at the two microphones, we have following relation:

$$
x_L(n) = h_{rel}(n) * x_R(n) + \underbrace{[v_L(n) - h_{rel}(n) * v_R(n))}_{v}.
\tag{12}
$$

The main goal is to estimate $h_{rel}$ given $x_L$ and $x_R$. The issue is in the presence of noise $v$ the estimation becomes challenging. The oracle ReIR is given as $h_{rel} = h_L * h_R^{-1}$, where $h_R^{-1}$ denotes the filter inverse to $h_R$. To ensure that the solution is causal, a fixed delay of a few milliseconds can be introduced (Lin et al., 2007; Koldovskỳ et al., 2013), i.e., $h_{rel} = h_L * h_R^{-1} * \delta(n - D)$, where $\delta(\cdot)$ is the unit impulse function and $D$ is the delay in samples. In the STFT domain, (12) can be equivalent to:

$$
X_L(f, t) = H_{rel}(f) X_R(f, t) + V(f, t),
\tag{13}
$$

where the oracle RTF is given by:

$$
H_{rel}(f) = \frac{H_L(f)}{H_R(f)}.
\tag{14}
$$

The goal again is to estimate $H_{rel}$ given the microphone signals $X_L$ and $X_R$. Note that under the assumption that the source position is fixed, we have the transfer functions independent of the time (thus omitting the index $t$).

## D.2 METHODS

**Signal processing-based methods**

- **Frequency-domain approaches:** The RTF estimation problem can be addressed in the frequency domain by utilizing signal and noise statistics, i.e., using the power spectral density (PSD) entities. To deal with noise, Gannot et al. (2001) propose the non-stationarity based frequency domain (NSFD) method relying on the assumption that noise signals are stationary, or less dynamic, when compared to the target speech signal. To be more exact, let $\Phi_{AB}(f, t)$ denote the (cross)-PSD between $A$ and $B$ during the $t$-th frame, we have:

$$\Phi_{X_L X_R}(f,t) = H_{rel}(f)\Phi_{X_R X_R}(f,t) + \Phi_{V X_R}(f,t). \tag{15}$$

Assuming over $t = 1, \ldots, P$ frames the the noise is stationary, we can write $\Phi_{V X_R}(f,t) = \Phi_{V X_R}(f)$ and solve the overdetermined set of equations:

$$\begin{bmatrix} \Phi_{X_L X_R}(f,1) \\ \vdots \\ \Phi_{X_L X_R}(f,P) \end{bmatrix} = \begin{bmatrix} \Phi_{X_R X_R}(f,1) & 1 \\ \vdots & \vdots \\ \Phi_{X_R X_R}(f,P) & 1 \end{bmatrix} \begin{bmatrix} H_{rel}(f) \\ \Phi_{V X_R}(f) \end{bmatrix}. \tag{16}$$

In practice the PSDs in the above set of equations are replaced by their sample estimates.

- **Time-domain approaches:** The ReIR estimation problem can be formulated in the time domain as: $\mathbf{h}_{rel} = \arg\min_{\mathbf{h} \in \mathbb{R}^M} \mathcal{L}(\mathbf{x}_L, \mathbf{X}_R \mathbf{h})$, where $\mathbf{x}_L = [x_L(1 - D), \ldots, x_L(N - D)]^T$ and $D$ is an integer delay for causality, and $\mathbf{X}_R$ is the convolution matrix of dimensions $N \times M$ constructed from $\mathbf{x}_R$. When the cost function $\mathcal{L}(\cdot, \cdot)$ is the squared Euclidean distance, the problem can be solved by least squares (noise-free) or regularized least squares (noisy) methods. To better handle the noise, prior knowledge about the ReIR can be leveraged to improve the estimation, e.g., utilizing the structural sparsity of ReIRs. One popular approach is the weighted $\ell_1$ approach (Benichoux et al., 2014; Koldovský et al., 2015; Giri et al., 2018):

$$\mathbf{h}_{rel} = \arg\min_{\mathbf{h} \in \mathbb{R}^M} \|\mathbf{x}_L - \mathbf{X}_R \mathbf{h}\|_2^2 + \lambda \|\mathbf{w} \odot \mathbf{h}\|_1, \tag{17}$$

where $\mathbf{w} = [w_1, \ldots, w_M]^T$ is a vector of non-negative weights and $\odot$ denotes the Hadamard product. To mimic the expected structure of a ReIR, the weights are chosen as follows:

$$w_i = k_1 e^{k_2 |i - D|^{k_3}}, i = 1, \ldots, M, \tag{18}$$

where $k_1$, $k_2$, and $k_3$ are positive constants and $D$ is the integer delay.

**Deep learning-based approach (proposed Align Net):** In the proposed AFnet, the first half Align Net is utilized to perform signal alignment by estimating RTFs, which can be viewed as a deep learning-based RTF estimation method. By exploiting the power of complex-valued deep networks for predicting the RTFs, the data-driven Align Net approach can be shown to outperform conventional model-based signal processing methods under noisy environments.

## D.3 EXPERIMENTS

**Settings:** We generate 2-mic noisy data using Pyroomacoustics library for conducting the RTF estimation experiments. We create a room of size $8 \times 8 \times 3$ (length×width×height in meters). Reverberation time is randomly chosen from $\{0.16, 0.32, 0.48, 0.64\}$ second. The simulating approach for estimating RIRs is based on the image source method. The two microphones are placed with a distance of 15 cm in between, where the center is placed at a distance of 1 meter to one of the walls, with a height of 1.525 meters, and the array is positioned horizontally to the wall with equal distance to the two sides. Speech and noise signals are randomly positioned at a distance of $\{1, 2, 3, 4\}$ meters from the microphone array. 649 files from the training set of AVSpeech dataset are used for training the Align Net and 160 files from the testing set are used for evaluating the competing

algorithms (i.e., Align Net, time-domain weighted $\ell_1$ approach, and NSFD approach). 4 types of commonly seen interference (blender, vacuum, washer, baby cry) are used for training and another 4 types (dog barking, kids playing sound, hair dryer, food sizzling) are used for testing. Each type of noise is multiplied with a scale randomly chosen from {0, 0.5, 1, 1.5, 2} before added up, and the combined noise is then added to the clean speech according to a specific SNR level randomly selected from {-10, -6, -3, 0, 3, 6, 10} dB for training and {0, 5, 10, 15} dB for testing.

For the freqency-domain approaches (Align Net and NSFD) we use the Hann window with 512-point FFT and a hop size of 256 for the STFT. For Align Net we use $C_1, C_2, C_3, C_4 = 32, 64, 64, 64$ and train for the RTF loss $\mathcal{L}_{\text{rtf}}$. For NSDF we use 250 frames as a processing block and smoothing over 5 frames for computing the PSD entities. For the time-domain weighted $\ell_1$ we use $k_1 = 0.1$, $k_2 = 0.11$ and $k_3 = 0.3$ with $\lambda = 0.0001$; and a ReIR length of $M = 1024$ for each processing block of 2048 samples.

**Evaluation metric:** To quantitatively evaluate the competing algorithms, we use a well-known and widely used performance metric called the Attenuation Rate (ATR) (Koldovskỳ et al., 2015), which can be evaluated as the ratio between $SNR_{out}$ and $SNR_{in}$ in dB scale, where:

$$SNR_{in} = \frac{\sum_{i \in \{L,R\}} \sum_n [h_i(n) * s(n)]^2}{\sum_{i \in \{L,R\}} \sum_n [v_i(n)]^2} \tag{19}$$

and

$$SNR_{out} = \frac{\sum_n [h_{rel}(n) * s_R(n) - s_L(n)]^2}{\sum_n [h_{rel}(n) * v_R(n) - v_L(n)]^2} \tag{20}$$

The numerator of $SNR_{out}$ measures the leakage of the target signal whereas the denominator measures the attenuation of the noise signal. The more negative the value (in dB) of ATR is, the better the evaluated algorithm performs.

**Results:** The ATR results are presented in Table 9. It can be seen that for all the methods the performance degrades as SNR decreases, since it becomes more challenging to estimate the RTF when noise is stronger. However, the proposed Align Net significantly outperforms the conventional model-based approaches over all SNR settings, indicating the effectiveness of the data-driven approach for the RTF estimation task against existing methods. In future study, improving the design of Align Net for better RTF estimation as well as its applications to other audio processing tasks (e.g., sound localization, speech separation, etc.) can be interesting topics to explore.

Table 9: Comparison of ATR (in dB) of proposed Align Net vs. conventional signal processing-based methods. The Align Net gives much better estimation outcomes for all the SNR settings.

| Methods | Input SNR | | | |
|---|---|---|---|---|
| | 0 dB | 5 dB | 10 dB | 15 dB |
| Unprocessed | -1.33 | -1.34 | -1.20 | -1.64 |
| NSFD | -2.79 | -3.79 | -5.14 | -5.78 |
| Weighted $\ell_1$ | -2.15 | -3.99 | -5.86 | -6.64 |
| Align Net (proposed) | -5.00 | -6.27 | -7.79 | -8.63 |

