# OpenReview forum: "Exploiting Spatial Separability for Deep Learning Multichannel Speech Enhancement with an Align-and-Filter Network"
_ICLR.cc/2023/Conference — Submitted to ICLR 2023_

### Official Review · Reviewer_6rDM · 2022-10-23

**Confidence:** 3
**Correctness:** 4
**Technical Novelty And Significance:** 2
**Empirical Novelty And Significance:** 2
**Recommendation:** 6

**Clarity, Quality, Novelty And Reproducibility:**

This work is similar to previous work (2 of the 3 refs. are not cited in the current paper) and it corresponds to a slight modifications of the Yoyama et al, 2020 paper.

-KOYAMA, Yuichiro et RAJ, Bhiksha. Exploring Optimal DNN Architecture for End-to-End Beamformers Based on Time-frequency References. arXiv preprint arXiv:2005.12683, 2020.
-KIM, Hansol, KANG, Kyeongmuk, et SHIN, Jong Won. Factorized MVDR Deep Beamforming for Multi-Channel Speech Enhancement. IEEE Signal Processing Letters, 2022, vol. 29, p. 1898-1902.
-KOYAMA, Yuichiro et RAJ, Bhiksha. Exploring Optimal DNN Architecture for End-to-End Beamformers Based on Time-frequency References. arXiv preprint arXiv:2005.12683, 2020.

 Despite the better results in comparison to the state of the art, the authors should further justify their changes in the proposed method architecture through a theoretical study, ablation study, etc.

To my opinion, this is the main weakness of this paper.



**Strength And Weaknesses:**

Strength:
-well written paper
-reproducible results (code and dataset)
-technically correct
-Convincing results


Weaknesses
-Not very original method

**Summary Of The Paper:**

This paper introduces a new deep-learning-based method for multichannel speech enhancement.
The proposed method so called AFnet uses a W-net 2D architecture which considers as the input a multi-channel time-frequency representation and  which predict both the alignment mask and the filtering mask which minimize the reconstruction error.

**Summary Of The Review:**

This is a well-written paper with reproducible results and a suitable evaluation methodology with freely available codes and dataset.

However, the proposed work is only an incremental contribution of a previously published work from Yoyama et al, 2020.

The authors could provide more arguments to explain in what their contribution is significant in comparison to the state of the art from a  theoretical point of view.

---

> ### Author Response · Authors · 2022-11-17
> **Response to Reviewer 6rDM**
>
> We would like to thank Reviewer 6rDM for the thoughtful comments and efforts towards improving our manuscript. Hope our following replies help answer the reviewer's outstanding questions.
>
> > **This work is similar to previous work (2 of the 3 refs. are not cited in the current paper) and it corresponds to a slight modifications of the Yoyama et al, 2020 paper.
> > Despite the better results in comparison to the state of the art, the authors should further justify their changes in the proposed method architecture through a theoretical study, ablation study, etc.
> To my opinion, this is the main weakness of this paper.**
>
> We thank the reviewer for mentioning the missing related works.
>
> As we describe our motivation in [Response to all reviewers](https://openreview.net/forum?id=DQou0RiwkR0&noteId=x3VOtYNQY07), phase alignment is an essential step for the two-stage neural network to better remove unwanted noise for enhancing speech. Although two-stage neural network designs have been proposed and motivated by the signal processing concept, till now most related works are not fully handling the spatial alignment problem based on exploiting the spatial signal properties among multi-microphone channel data.
>
> Thus, the main contribution of our work is not the adoption of W-Net, but the way to joint the two stages (i.e., the two U-Net) for more efficiently performing spatial and spectral filtering of the noisy multichannel speech. In previous deep learning-based multichannel SE works, such as [Wang et al., 2021], the spatial information has to be implicitly learned by utilizing conventional signal processing-based beamformer units such as MVDR or multichannel Wiener Filter (MWF). However, integrating the conventional beamformers would require several matrix operations such as matrix inversion or eigendecomposition that could cause the computational cost to be high when there are more microphones along with instability issues, making scaling up a more challenging task; otherwise special cares have to be taken to avoid so such as [H. Kim et al., 2022]. Our main novelty lies in the proposed RTF supervised learning of the alignment masks, which can be achieved by simply adding skip connections in between the U-Net modules, to explicitly incorporate spatial information into model learning, thereby bypassing the need of involving conventional beamformer units while exploiting useful and interpretable spatial features within multichannel data.
>
> + [Wang et al., 2021] Z.-Q. Wang et al. "Sequential multi-frame neural beamforming for speech separation and enhancement," in _IEEE Spoken Language Technology Workshop (SLT)_, 2021.
>
> + [H. Kim et al., 2022] H. Kim et al., “Factorized MVDR Deep Beamforming for Multi-Channel Speech Enhancement,” _IEEE Signal Processing Letters_, 2022.
>
> To differentiate from the W-Net beamformers of [Koyama & Raj, 2019; Koyama & Raj, 2020], we first point out that their approaches utilize a multiple-input-single-output (MISO) module followed by a multiple-input-multiple-output (MIMO) module, while in our AFnet the two stages are both MIMO. To be more specific, the major difference is in that the first U-Net in their works aims to predict a single-channel enhanced output as a “reference signal” to the second stage, while in our AFnet it estimates multiple masks for predicting aligned multichannel signals to exploit spatial information across all channels. Therefore, the role of the first U-Net is different, where in their works it is used to summarize the multiple noisy input into a cleaner single-channel representation, while in our AFnet it dedicates to leveraging spatial information of the microphone channels to reduce the burden of the subsequent filtering module. In addition, since our approach is highly dependent on the phase components of RTFs, we have adopted complex-valued networks to take advantage of the complex nature of STFTs, while in their works the networks only utilize real-valued operations.
>
> + [Koyama & Raj, 2019] Y. Koyama and B. Raj, “W-Net BF: DNN-based beamformer using joint training approach,” arXiv:1910.14262, 2019.
>
> + [Koyama & Raj, 2020] Y. Koyama and B. Raj, “Exploring optimal DNN architecture for end-to-end beamformers based on time-frequency references,” arXiv:2005.12683, 2020.
>
> We have also conducted much more evaluations in our revised manuscript to support the effectiveness and robustness of the AFnet for unseen room configurations, time-varying speaker locations, etc. We hope the additional discussion in *Section 4.2.6: Are intermediate alignment masks necessary?, Appendix C.1: Comparison of RTF-aware training schemes (two-step vs. regularized), Appendix C.2: Fixing vs. unfixing Align Net while minimizing reconstruction loss, Appendix C.5: Generalization to other room configurations, Appendix C.6: SE under time-varying RIR scenarios* of the revised manuscript will help strengthen the novelty and further justify the contribution of our work.

---

> ### Author Response · Authors · 2022-11-30
> **Available for Discussion**
>
> We hope our responses have addressed the reviewer's concerns, but if not we are available/open to further address any outstanding issues.

---

> ### Author Response · Authors · 2022-12-06
> **Additional experiments and results available**
>
> Dear Reviewer 6rDM,
>
> Based on the suggestions and feedback from another reviewer, we have conducted further experiments and gathered more evidence to support our proposed AFnet method. We think you might be interested in taking a look at [here](https://openreview.net/forum?id=DQou0RiwkR0&noteId=Q-hdG5es2p9) and your feedback is welcome. We hope you find the additional results also interesting; and, reinforce your positive view of the paper and make you more supportive of our work.
>
> Sincerely,
>
> Authors of Paper1144

---

### Official Review · Reviewer_jGjP · 2022-10-24

**Confidence:** 5
**Correctness:** 3
**Technical Novelty And Significance:** 3
**Empirical Novelty And Significance:** 3
**Recommendation:** 6

**Clarity, Quality, Novelty And Reproducibility:**

The paper is relatively well written, with some typos and inconsistencies.
The novelty is relatively small, but sufficient and relevant.
Appendix has several relevant results, which should be included in the main text.
Components of the datasets are publicly available. However, training details seem to be omitted, and it may be relatively difficult to reproduce the exact results without the access to the code for data preparation.

**Strength And Weaknesses:**

I think the basic idea is sensible and relevant: adding the RTF-based loss to improve construction of a multi-channel filter. The presented results indicate that this has a considerable effect on the proposed method.

Some weaknesses which should be addressed
- "To the best of our knowledge, the only work that incorporates RTFs for SE is by Wang & Wang (2018), in which, however, the RTF estimation is used as an intermediate step to assist the conventional MVDR beamformer only." -- Many DNN-based speech enhancement papers estimate RTFs for speech enhancement. For example, "ADL-MVDR: All deep learning MVDR beamformer for target speech separation" is a relatively recent example which estimates a steering vector, which is an RTF. It is true that, in this instance, the vector is used to estimate the filter.
- Furthermore, very relevant work is the following paper: The PCG-AIID System for L3DAS22 Challenge (https://arxiv.org/abs/2202.10017). The system used there is using a structure very similar to the one used here (W-like, multichannel intermediate output and single-channel final output).
- I think it should be clarified that the AlignNet is not doing only temporal alignment, but also level adjustment. Even more critical, the results from B.1 should be included in the main text. One of the main questions for me was if the magnitude equalization is really important for this work, and I expected the phase to be the important part. B.1 clearly gives the answer: phase adjustment is sufficient. It's therefore not clear why is this not included in the main text.
- The above brings another questions. If phase alignment is enough, would it be suffice to have a phase alignment loss only for the final output mask, without using the intermediate mask? I would assume that this would be sufficient, and that having alignment in the middle of the W structure is not necessarily critical. However, it would be good to get clarification from the authors, and to understand if the two-stage structure is critical (or it's suffice to have a skip connection and a combined loss at the output).
- Leaky ReLU is used for real and imaginary components independently. However, the slope is not defined anywhere in the paper. This is interesting, since it may mean the complex gain values are relatively strongly pushed to be in the first quadrant, which is not very intuitive.
- Table 1 -- These results indicate that the proposed FilterNet fails on the original unaligned signals. Operating on aligned signals is expected to perform better, but FilterNet on unaligned signals fails completely. This is quite surprising, since multi-channel systems proposed in other papers work well without the two-stage process (align+filter). I'm curious if this failure could be related to the choice of nonlinearity (above).
- Section 4.2.2 / Table 2 -- It would be interesting to connect these results to Table 1, e.g., what's the gap of the AFnet compared to perfectly adjusted signals from Table 1.
- Some typos, e.g., "Nerual BF"


**Summary Of The Paper:**

The paper presents an align-and-filter structure for neural network-based multi-channel speech enhancement.
The paper claims that the proposed decomposition of a filter into an element-wise product of alignment gain and a filtering gain is critical for an improved performance. Furthermore, it is claimed that training with an RTF loss is critical for the success of the proposed method.

**Summary Of The Review:**

The paper claims that the proposed two-stage network with an RTF-based loss is a better alternative to single-stage multi-channel processing systems. Experimental results are promising.
However, there are several open questions which need to be addressed before publishing the paper.

---

The authors addresses most of my comments in their responses, and I've updated the score to 6.

---

> ### Author Response · Authors · 2022-11-17
> **Response to Reviewer jGjP (1/3)**
>
> We would like to thank Reviewer jGjP for the thoughtful comments and efforts towards improving our manuscript. Hope our following replies help answer the reviewer's outstanding questions.
>
> > **To the best of our knowledge, the only work that incorporates RTFs for SE is by Wang & Wang (2018), in which, however, the RTF estimation is used as an intermediate step to assist the conventional MVDR beamformer only." -- Many DNN-based speech enhancement papers estimate RTFs for speech enhancement. For example, "ADL-MVDR: All deep learning MVDR beamformer for target speech separation" is a relatively recent example which estimates a steering vector, which is an RTF. It is true that, in this instance, the vector is used to estimate the filter.**
>
> Thank you for pointing out other relevant papers to the RTF estimation. We have mentioned these papers in related work of the revised manuscript that also utilizes RTF in the SE model and modified the corresponding statement to “However, the utilization of such important RTF information is often overlooked in the model design of DNN-based multichannel SE. Although several works have incorporated RTFs for SE (Wang & Wang, 2018; Zhang et al., 2021), the RTF estimation is mostly used as an intermediate step to assist the MVDR beamformer only.”
>
> > **Furthermore, very relevant work is the following paper: The PCG-AIID System for L3DAS22 Challenge ([https://arxiv.org/abs/2202.10017](https://arxiv.org/abs/2202.10017)). The system used there is using a structure very similar to the one used here (W-like, multichannel intermediate output and single-channel final output).**
>
> We thank the reviewer for putting this work in the picture. We are aware that our AFnet shares similarity with several two-stage networks that also features a W-Net architecture, e.g., [Wang et al., 2021; Koyama & Raj, 2019; Koyama & Raj, 2020]. Please refer to [Response to all reviewers](https://openreview.net/forum?id=DQou0RiwkR0&noteId=x3VOtYNQY07) for the discussion on the differentiation with existing works.
>
> + [Wang et al., 2021] Z.-Q. Wang et al. "Sequential multi-frame neural beamforming for speech separation and enhancement," in _IEEE Spoken Language Technology Workshop (SLT)_, 2021.
>
> + [Zhang et al., 2021] Z. Zhang et al., “ADL-MVDR: All deep learning MVDR beamformer for target speech separation,” in _IEEE International Conference on Acoustics, Speech, and Signal Processing (ICASSP)_, 2021.
>
> + [Koyama & Raj, 2019] Y. Koyama and B. Raj, “W-Net BF: DNN-based beamformer using joint training approach,” _arXiv:1910.14262_, 2019.
>
> + [Koyama & Raj, 2020] Y. Koyama and B. Raj, “Exploring optimal DNN architecture for end-to-end beamformers based on time-frequency references,” _arXiv:2005.12683_, 2020.
>
> > **I think it should be clarified that the AlignNet is not doing only temporal alignment, but also level adjustment. Even more critical, the results from B.1 should be included in the main text. One of the main questions for me was if the magnitude equalization is really important for this work, and I expected the phase to be the important part. B.1 clearly gives the answer: phase adjustment is sufficient. It's therefore not clear why is this not included in the main text.**
>
> We are grateful for the reviewer to bring up this important observation regarding the importance of phase alignment. We agree with the reviewer that this is an important aspect to be highlighted in the main text instead of placed in Appendix. Yes, spatial features of sound sources highly correlate with the phase components as they reflect the time delay in the temporal domain, and therefore are more essential for locating the target speech. We have moved the related discussion from Appendix to _Section 4.2.5: Importance of phase alignment in RTF estimation_ in the revised manuscript and noted that Align Net performs both temporal and level alignments.

---

> > ### Author Response · Authors · 2022-11-17
> > **Response to Reviewer jGjP (2/3)**
> >
> > > **The above brings another questions. If phase alignment is enough, would it be suffice to have a phase alignment loss only for the final output mask, without using the intermediate mask? I would assume that this would be sufficient, and that having alignment in the middle of the W structure is not necessarily critical. However, it would be good to get clarification from the authors, and to understand if the two-stage structure is critical (or it's suffice to have a skip connection and a combined loss at the output).**
> >
> > Thank you for bringing up this question which we also think is interesting and worth looking into. We have studied such a case where an RTF phase alignment loss is directly applied to the output filter weights (i.e., filtering masks) of the W-Net without having the intermediate masks. The results and discussion are presented in Table 4 of _Section 4.2.6: Are intermediate alignment masks necessary?_ of the revised manuscript (and also presented in the below table), where we observe that the phase regularization could slightly improve the performance of the W-Net but is still far from matching the performance of the AFnet that utilizes RTF supervision on the intermediate masks. From there, we conclude that the two-stage structure is critical.
> >
> > Table: Results of AFnet and W-Net with RTF phase regularization on the filtering masks (taken from Table 4 of the revised manuscript).
> >
> > |                  |  | PESQ | | | STOI | | | SSNR ||
> > |-------------------------------------------------------|--------------------------|--------------------------|--------------------------|--------------------------|--------------------------|--------------------------|--------------------------|--------------------------|--------------------------|
> > | Method                              | 2-mic | 4-mic | 8-mic | 2-mic | 4-mic | 8-mic | 2-mic | 4-mic | 8-mic |
> > | W-Net for filter weights, no phase reg.               | 1.74                     | 1.61                     | 1.67     | 0.703 | 0.679 | 0.688 | 4.80 | 4.07 | 4.39
> > | W-Net for filter weights, phase reg. $\lambda=0.0001$ | 1.76                     | 1.61                     | 1.65 | 0.713 | 0.686 | 0.692 | 4.95 | 4.21 | 4.42
> > | W-Net for filter weights, phase reg. $\lambda=0.001$  | 1.63                     | 1.60                     | 1.68  | 0.688 | 0.686 | 0.702 | 4.36 | 4.14 | 4.60
> > | W-Net for filter weights, phase reg. $\lambda=0.01$   | 1.71                     | 1.59                     | 1.61   | 0.705 | 0.683 | 0.684 | 4.55 | 4.14 | 4.30
> > | W-Net for filter weights, phase reg. $\lambda=0.1$    | 1.73                     | 1.66                     | 1.74    | 0.708 | 0.701 | 0.710 | 4.09 | 3.54 | 3.87
> > | AFnet                                                 | 1.84                     | 1.92                     | 1.99                  | 0.728 | 0.739 | 0.753 | 5.06 | 5.26 | 5.35
> >
> >
> > > **Leaky ReLU is used for real and imaginary components independently. However, the slope is not defined anywhere in the paper. This is interesting, since it may mean the complex gain values are relatively strongly pushed to be in the first quadrant, which is not very intuitive.**
> >
> > Thank you for recommending to indicate the slope of the leaky ReLU. The slope was set to 0.01 in our experiments and we have noted it in the revised manuscript. In fact, the utilization of leaky ReLU for the real and imaginary parts follows the work of [Choi et al., 2018]. We have not seen any discussion on the impact of the slope in their paper so we are not sure if leaky ReLU will cause certain issues. But since the final output layer of the U-Net model in our network has no nonlinearity, it seems that the estimated masks will not be pushed to the first quadrant at least for the output gain values.
> >
> > + [Choi et al., 2018] H.-S. Choi et al., “Phase-aware speech enhancement with deep complex U-Net,” _arXiv:1903.03107_, 2018.

---

> > > ### Author Response · Authors · 2022-11-17
> > > **Response to Reviewer jGjP (3/3)**
> > >
> > > > **Table 1 -- These results indicate that the proposed FilterNet fails on the original unaligned signals. Operating on aligned signals is expected to perform better, but FilterNet on unaligned signals fails completely. This is quite surprising, since multi-channel systems proposed in other papers work well without the two-stage process (align+filter). I'm curious if this failure could be related to the choice of nonlinearity (above).**
> > >
> > > The results in Table 1 actually might not have indicated that the Filter Net on unaligned signals fails completely, but only that it performs inferior to the case on the aligned signals. Note that the Filter Net is still able to denoise the noisy inputs for the unaligned case, as the PESQ, STOI, and SSNR values in Table 1 are higher than the noisy numbers in Table 2 (Noisy: PESQ: 1.21 / STOI: 0.577 / SSNR: -1.70). We do not think such an observation is related to the nonlinearity, as for the aligned signal case the model still uses the same nonlinearity but works well. Instead, we think the difference might be mainly attributed to whether alignment is performed or not rather than the nonlinearity used.
> > >
> > > > **Section 4.2.2 / Table 2 -- It would be interesting to connect these results to Table 1, e.g., what's the gap of the AFnet compared to perfectly adjusted signals from Table 1.**
> > >
> > > Thank you for the suggestion. We have mentioned this aspect in _Section 4.2.4: Comparison to SOTA methods_ of the revised manuscript: “Note that the performance of AFnet lies in between the aligned and unaligned signal cases in Table 1, which is reasonable as the alignment is never perfect but only to certain degree.”
> > >
> > > > **Some typos, e.g., "Nerual BF"**
> > >
> > > Thank you for pointing out the typos. We have corrected them in the revised manuscript.
> > >
> > > > **The paper is relatively well written, with some typos and inconsistencies. The novelty is relatively small, but sufficient and relevant. Appendix has several relevant results, which should be included in the main text. Components of the datasets are publicly available. However, training details seem to be omitted, and it may be relatively difficult to reproduce the exact results without the access to the code for data preparation.**
> > >
> > > We have corrected typos at our best in the revised manuscript and moved relevant results originally in the Appendix to the main text as suggested by the reviewer. We agreed that the RTF phase alignment is an important aspect to highlight, which also leads to further discussion on potential phase regularization following up. We have also included more details regarding model training and data preparation. We hope that it would mitigate the difficulty of reproducing the results in the paper.

---

> > > > ### Comment · Reviewer_jGjP · 2022-12-01
> > > > **Thank you for your responses**
> > > >
> > > > I would like to thank the authors for their responses.
> > > > My comments have been mostly addressed and the paper has been improved.

---

> > > > > ### Author Response · Authors · 2022-12-02
> > > > > **Thanks for increasing the score and for all the insightful suggestions to help improve the paper**
> > > > >
> > > > > Dear Reviewer jGjP,
> > > > >
> > > > > We appreciate very much that you have increased the score after we improved the paper based on your comments and suggestions!
> > > > >
> > > > >
> > > > > Sincerely,
> > > > >
> > > > > Authors of Paper1144

---

> ### Author Response · Authors · 2022-11-30
> **Available for Discussion**
>
> We hope our responses have addressed the reviewer's concerns, but if not we are available/open to further address any outstanding issues.

---

> ### Author Response · Authors · 2022-12-02
> **Thanks for increasing the score and for all the insightful suggestions to help improve the paper**
>
> Dear Reviewer jGjP,
>
> We appreciate very much that you have increased the score after we improved the paper based on your suggestions!
>
> Sincerely,
>
> Authors of Paper1144

---

> ### Author Response · Authors · 2022-12-06
> **Additional experiments and results available**
>
> Dear Reviewer jGjP,
>
> Based on the suggestions and feedback from another reviewer, we have conducted further experiments and gathered more evidence to support our proposed AFnet method. We think you might be interested in taking a look at [here](https://openreview.net/forum?id=DQou0RiwkR0&noteId=Q-hdG5es2p9) and your feedback is welcome. We hope you find the additional results also interesting; and, reinforce your positive view of the paper and make you more supportive of our work.
>
> Sincerely,
>
> Authors of Paper1144

---

### Official Review · Reviewer_aM96 · 2022-10-24

**Confidence:** 4
**Clarity, Quality, Novelty And Reproducibility:** Not novel enough or at par with the s…
**Correctness:** 2
**Technical Novelty And Significance:** 2
**Empirical Novelty And Significance:** 2
**Recommendation:** 3

**Strength And Weaknesses:**

The Paper is very hard to read, where I found lots of details, equations, symbols being intermingled within inline text.

- I couldn't find the motivation behind the proposed framework. Why the choices are made, how they impact the learning process, and what extra they provide over existing studies.
- This is another classical example of an empirical deep-learning paper where authors have tried to combine various existing ideas without any theoretical backing.
- The link between speech production/perception as well as multichannel setting with the choices made is clearly missing.

- Authors mentioned that their work is [Inspired by the alignment concept in signal processing]. What is this alignment process? no reference is provided, neither any discussion on why this is important.
- The alignment and filter network are very similar. Why not just train a single big network with the same capacity?
  In experiments, authors have shown results with individual networks empirically, but it is unclear if the model capacity is similar or not.
-  ILD and ITD correspondence to the magnitude and phase spectra is exploited using a complex neural network.
While working in complex STFT domain using conventional methods is well understood, the same is not the case with DNNs. In-fact their are a many works which have shown that the same task can be performed entirely in time domain by using learned filterbanks modelling raw waveform directly in an end-to-end fashion.
Complex networks are very difficult to optimize in general, especially for speech/audio tasks.
- Authors are suggested to avoid MSE loss as existing works have highlighted the drawbacks and shown better performance by using STOI or its variants for enhancement tasks.
- I fail to understand the rationale behind split training. Why not joint training right from scratch? May be some visualization based on the loss landscape of the overall model (in complex and real settings) and what the network is learning in terms of filter responses/saliency maps or geometric properties of network weights is highly encouraged for explainability.


Experiments:
- lots of details are missing, which makes it very hard to reproduce the experiments
- given the empirical nature of the paper, Authors should submit the code and pre-trained models.
- details about existing works with which comparison is made are missing. Why these were chosen? I only see FaSNet closely related to this work.

**Summary Of The Paper:**

This work proposes an autoencoder based multi-channel speech enhancement framework. Overall the work is applied and mostly empirical in nature. The proposed framework is based on existing well-established ideas from empirical deep learning. There are no theoretical insights and the contribution in terms of the combination of existing ideas is relatively marginal.

**Summary Of The Review:**

While the problem addressed is interesting, the proposed ideas are just incremental and not novel enough for the ICLR main conference.
Authors are recommended to resubmit to a suitable workshop, but the question about the theoretical contribution is the main weakness of the paper.

---

> ### Author Response · Authors · 2022-11-17
> **Response to Reviewer aM96 (1/4)**
>
> We would like to thank Reviewer aM96 for the thoughtful comments and efforts towards improving our manuscript. Hope our following replies help answer the reviewer's outstanding questions.
>
> >**The Paper is very hard to read, where I found lots of details, equations, symbols being intermingled within inline text.**
>
> We have optimized our main proposed system depicted in Figure 1 to make it more self-contained. We have moved several inline but important equations to be stand-alone. We have also modified the notations, e.g., using bold, capital, non-italic letters for matrices and normal, capital, italic letters for their entries following the signal processing convention for the equations describing the proposed approach. We have also included a summary of the proposed training scheme in Algorithm 1 in *Appendix B.1: RTF-aware training algorithms* of the revised manuscript. We sincerely hope that these changes help improve the reading of the work and understanding of the proposed approach.
>
> >**I couldn't find the motivation behind the proposed framework. Why the choices are made, how they impact the learning process, and what extra they provide over existing studies.**
>
> The alignment followed by filtering process is a well-known design concept that has been used in the signal processing field as seen in reference [Gannot et al., 2001; Cohen, 2004; Krueger et al., 2010; Koldovsky et al., 2015]. The benefit of using a two-stage framework is that by dividing the SE task into the subtasks of learning to align and learning to filter leads to a more efficient and interpretable enhancement process that explicitly incorporates multichannel data. It is known that signal alignment is beneficial for boosting the SNR through constructive combination of the target signals. Our network design incorporates such aspect and comes out in the form of a two-stage model. For more details, please refer to *Section 4.2.2: Exploiting spatial separability with AFnet* of the revised manuscript which gives insights on  how the two-stage neural network can benefit multichannel SE compared to one-stage neural networks.
> + [Gannot et al., 2001] S. Gannot et al., “Signal enhancement using beamforming and nonstationarity with applications to speech,” *IEEE Transactions on Signal Processing,* 2001.
>
>  + [Cohen, 2004] I. Cohen, “Relative transfer function identification using speech signals,” *IEEE Transactions on Speech and - Audio Processing,* 2004.
>
> + [Krueger et al., 2010] A. Krueger et al., “Speech enhancement with a GSC-like structure employing eigenvector-based transfer function ratios estimation,” *IEEE Transactions on Audio, Speech, and Language Processing,* 2010.
>
> + [Koldovsky et al., 2015] Z. Koldovsky et al, “Spatial source subtraction based on incomplete measurements of relative transfer function,” *IEEE/ACM Transactions on Audio, Speech, and Language Processing,* 2015.
>
> However, most current two-stage neural network designs have yet to adopt such design concept efficiently. For instance, some related works aim to directly learn a final single channel (mono) clean speech output without implicitly or explicitly regularize the neural network to combine multiple microphones. Without taking care of the multichannel alignment aspect, the neural network might only rely on the single-channel output information and lack guidance on how to efficiently extract and combine multichannel spatial characteristics. This is also the missing part for all the previous related works which has driven our motivation of incorporating RTFs.
>
> We summarize our contributions below to show how the related works compare to ours and the experimental results to validate our proposed method.
>
> - **One-stage vs. two-stage:**
> The advantage of the two-stage design incorporating the RTF alignment over typical one-stage SE networks is the better exploitation of spatial separability. Typical one-stage SE networks do not explicit learn the spatial features. Thus, they might have difficulty in simultaneously learning spatial enhancement (SNR improvement via combining multichannel signals) and spectral enhancement (frequency domain denoising). For more details please refer to *Section 4.2.2: Exploiting spatial separability with AFnet* of the revised manuscript.

---

> > ### Author Response · Authors · 2022-11-17
> > **Response to Reviewer aM96 (2/4)**
> >
> > - **Split vs. non-split training:**
> > The split training (i.e., first train Align Net then jointly train Align Net and Fitler Net) can first guide the network to learn how to align signals before actually learning how to filter. Another way of RTF-aware training is by combining the RTF loss and signal reconstruction loss together and jointly train the entire AFnet from scratch (non-split). Both approaches can incorporate spatial properties within the RTFs. However, we have empirically shown that the split training approach (Algorithm 1) performs consistently better than the non-split joint training approach (Algorithm 2) (see the table below which is taken from Table 5 of the revised manuscript). For more related discussion please refer to *Appendix C.1: Comparison of RTF-aware training schemes (two-step vs. regularized)* of the revised manuscript.
> >
> > Table: Results of AFnet trained by Algorithm 1 (split) and Algorithm 2 (non-split) RTF-aware training schemes (taken from Table 5 of the revised manuscript).
> > |                  |  | PESQ | | | STOI | | | SSNR ||
> > |-------------------------------------------------------|--------------------------|--------------------------|--------------------------|--------------------------|--------------------------|--------------------------|--------------------------|--------------------------|--------------------------|
> > | Method                              | 2-mic | 4-mic | 8-mic | 2-mic | 4-mic | 8-mic | 2-mic | 4-mic | 8-mic |
> > | Algorithm 1                     | 1.84                     | 1.92                     | 1.99                     | 0.728 | 0.739 | 0.753 | 5.06 | 5.26 | 5.35
> > | Algorithm 2, $\lambda=0$      | 1.80                     | 1.72                     | 1.88                     | 0.720 | 0.711 | 0.734 | 4.99 | 4.66 | 5.05
> > | Algorithm 2, $\lambda=0.0001$ | 1.77                     | 1.77                     | 1.88                     | 0.718 | 0.714 | 0.735 | 4.80 | 4.76 | 5.10
> > | Algorithm 2, $\lambda=0.001$  | 1.80                     | 1.73                     | 1.88                     | 0.721 | 0.714 | 0.735 | 4.98 | 4.64 | 5.10
> > | Algorithm 2, $\lambda=0.01$   | 1.78                     | 1.82                     | 1.84                     | 0.718 | 0.729 | 0.726 | 4.91 | 4.94 | 4.98
> > | Algorithm 2, $\lambda=0.1$    | 1.82                     | 1.82                     | 1.95                     | 0.723 | 0.726 | 0.744 | 5.09 | 4.94 | 5.28
> > | Algorithm 2, $\lambda=1$      | 1.76                     | 1.79                     | 1.88                     | 0.713 | 0.720 | 0.731 | 4.81 | 4.89 | 5.13
> >
> >
> > - **Training  with vs. without RTFs:**
> > It is of great importance to explicitly incorporate the spatial information for multichannel SE. Through RTF supervision, the AFnet is able to better capture the spatial characteristics for performing SE based on spatial separability. We have interpreted the effect of RTF supervision on the learning alignment masks and linked it to the spatial properties of sound sources as discussed in *Section 4.2.3: Visualizing the alignment outcomes* of the revised manuscript.
> >
> > > **This is another classical example of an empirical deep-learning paper where authors have tried to combine various existing ideas without any theoretical backing.**
> > >
> > We would like to argue that our approach is not a combination of existing ideas, but rather a transfer of conventional model-based signal processing wisdom to the data-driven deep learning area. Motivated by the align-then-filer concept from signal processing, the proposed AFnet approach via the RTF supervision with the two-stage design has not be seen before in deep learning-based systems. On the contrary, most existing SE systems leverage empirical deep learning techniques to directly learn the mono target speech audio without thoughtfully considering the signal spatial properties within multiple microphones. To support the motivation of this alignment design principle of the AFnet, we have included an additional section that details several well-known signal processing algorithms based on this design principle. Please kindly refer to *Appendix A: Conventional signal processing beamforming algorithms based on align-then-filter process* of the revised manuscript.
> >
> > > **The link between speech production/perception as well as multichannel setting with the choices made is clearly missing.**
> >
> > Our AFnet carries out the “align-then-filter” mechanism to enhance speech. This imitates the human listening behavior where people tend to face (steer) toward the target speaker (i.e., to temporally align the sound waveforms arriving at both ears) as to better perceive the speech. We have described this link between our AFnet and speech perception in the revised manuscript.

---

> > > ### Author Response · Authors · 2022-11-17
> > > **Response to Reviewer aM96 (3/4)**
> > >
> > > > **Authors mentioned that their work is [Inspired by the alignment concept in signal processing]. What is this alignment process? no reference is provided, neither any discussion on why this is important.**
> > >
> > > We feel sorry about that the reviewer did not spot the place where we mentioned the alignment concept in the Introduction section of the initial submission. To further enhance the description, we have added several relevant works in *Section 1: Introduction* of the revised manuscript to the place where we describe the alignment concept. We have also added a few words to the sentence “Inspired by the alignment concept in signal processing…” to link to the particular description. Moreover, we have included *Appendix A: Conventional signal processing beamforming algorithms based on align-then-filter process* in the revised manuscript to review several conventional beamforming algorithms that are based on the alignment aspect.
> > >
> > > > **The alignment and filter network are very similar. Why not just train a single big network with the same capacity? In experiments, authors have shown results with individual networks empirically, but it is unclear if the model capacity is similar or not.**
> > >
> > > Training a single, large network and letting it blindly learn to exploit meaningful spatial information of the multichannel data can be more challenging. That is the reason why some existing works integrate conventional model-based beamformer units (e.g., MVDR, multichannel Wiener filter (MWF)) into the large network to facilitate the learning of spatial features, e.g., [Wang et al., 2021; Zhang et al., 2021; H. Kim et al., 2022]. Our AFnet bypasses this step by incorporating RTF information during training to explicitly promote the learning of spatial features. In other words, the Align Net aims to perform spatial enhancement (microphone array processing) for capturing the signal coming only from a particular direction (of the speaker). The second stage Filter Net aims to combine the aligned signals by leveraging spectral features to perform further denoising. In our experiments, we have shown that a network with the same capacity would not be comparable to the AFnet (e.g., please kindly refer to Figures 2, 6, 7, 8 and Tables 4, 7, 8 of the revised manuscript by comparing W-Net with the AFnet which are of the same model capacity).
> > >
> > > + [Wang et al., 2021] Z.-Q. Wang et al. "Sequential multi-frame neural beamforming for speech separation and enhancement," in *IEEE Spoken Language Technology Workshop (SLT),* 2021.
> > >
> > > + [Zhang et al., 2021] Z. Zhang et al., “ADL-MVDR: All deep learning MVDR beamformer for target speech separation,” in *IEEE International Conference on Acoustics, Speech, and Signal Processing (ICASSP),* 2021.
> > >
> > > + [H. Kim et al., 2022] H. Kim et al., “Factorized MVDR Deep Beamforming for Multi-Channel Speech Enhancement,” *IEEE Signal Processing Letters,* 2022.
> > >
> > > > **ILD and ITD correspondence to the magnitude and phase spectra is exploited using a complex neural network. While working in complex STFT domain using conventional methods is well understood, the same is not the case with DNNs. In-fact their are a many works which have shown that the same task can be performed entirely in time domain by using learned filterbanks modelling raw waveform directly in an end-to-end fashion. Complex networks are very difficult to optimize in general, especially for speech/audio tasks.**
> > >
> > > We agree that there are many works directly operate in the waveform domain for SE. However, there are also many other works that demonstrate the benefit of utilizing STFT domain processing over the waveform domain processing for the SE task, especially in the case where phase is an important factor, e.g., [Choi et al., 2018; Tolooshams et al., 2020; Hu et al., 2020]. Our work follows these previous arts to improve the STFT-based SE approaches for multichannel data.
> > >
> > > + [Choi et al., 2018] H.-S. Choi et al., “Phase-aware speech enhancement with deep complex U-Net,” *arXiv:1903.03107,* 2018.
> > >
> > > + [Tolooshams et al., 2020] B. Tolooshams et al., “Channel-attention dense U-Net for multichannel speech enhancement,” in *IEEE International Conference on Acoustics, Speech, and Signal Processing (ICASSP),* 2020.
> > >
> > > + [Hu et al., 2020] Y. Hu et al., “DCCRN: Deep complex convolution recurrent network for phase-aware speech enhancement,” in *Annual Conference of the International Speech Communication Association (Interspeech),* 2020.

---

> > > > ### Author Response · Authors · 2022-11-17
> > > > **Response to Reviewer aM96 (4/4)**
> > > >
> > > > > **Authors are suggested to avoid MSE loss as existing works have highlighted the drawbacks and shown better performance by using STOI or its variants for enhancement tasks.**
> > > >
> > > > We would first clarify that the MSE loss is only applied to the RTF supervision but not to the final reconstructed signal. For the reconstructed signal we actually use the “combined power-law compressed MSE loss” which is a variant of the conventional MSE loss that has been widely adopted in many existing works, e.g., [Wilson et al., 2018; Ephrat et al., 2018; Fedorov et al., 2020], whose effectiveness is supported by the work of [Braun et al., 2021] which investigates and contrasts several loss functions for the SE task. STOI and other variants could also be used in our framework but the effect on the loss functions is beyond the scope of this paper.
> > > >
> > > > + [Wilson et al., 2018] K. Wilson et al., “Exploring tradeoffs in models for low-latency speech enhancement,” in *International Workshop on Acoustic Signal Enhancement (IWAENC),* 2018.
> > > >
> > > > + [Ephrat et al., 2018] A. Ephrat et al., “Looking to listen at the cocktail party: A speaker-independent audio-visual model for speech separation,” in *ACM Transactions on Graphics,* 2018.
> > > >
> > > > + [Fedorov et al., 2020] I. Fedorov et al., “TinyLSTMs: Efficient Neural Speech Enhancement for Hearing Aids,” in *Annual Conference of the International Speech Communication Association (Interspeech),* 2020.
> > > >
> > > > + [Braun et al., 2021] S. Braun et al., “A consolidated view of loss functions for supervised deep learning-based speech enhancement,” in *International Conference on Telecommunications and Signal Processing (TSP),* 2021.
> > > >
> > > >
> > > > > **I fail to understand the rationale behind split training. Why not joint training right from scratch? May be some visualization based on the loss landscape of the overall model (in complex and real settings) and what the network is learning in terms of filter responses/saliency maps or geometric properties of network weights is highly encouraged for explainability.**
> > > >
> > > > To answer on split training, we have added new results of Table 5 in the revised manuscript to show that joint training of the signal reconstruction loss and the alignment loss combined (in a regularized loss form) right from scratch could not match the performance of the AFnet utilizing the two-step (split-then-joint) training (i.e., train Align Net first then train the entire network together).
> > > >
> > > > Figures 3, 5, and 9 of the revised manuscript visualize the learned alignment masks of using the split-then-joint RTF-aware training and normal RTF-agnostic training schemes. We can see that with the RTF supervision it becomes more interpretable in terms of the learned intermediate variables associated with spatial diversity. To conclude, the AFnet by leveraging multichannel RTF information during training is more explainable than the model trained to minimize only the reconstruction loss on the mono target speech audio and performs more efficient enhancement due to exploiting spatial characteristics.
> > > >
> > > > > **lots of details are missing, which makes it very hard to reproduce the experiments**
> > > >
> > > > We have added more implementation details to *Appendix B: Implementation details* of the revised manuscript. We hope that the added details would be helpful for improving the reproducibility of the work.
> > > >
> > > > > **details about existing works with which comparison is made are missing. Why these were chosen? I only see FaSNet closely related to this work.**
> > > >
> > > > We have detailed the related settings of existing works that we compare in *Appendix B: Implementation details* of the revised manuscript. For the models in Table 2, we have chosen them as to compare single (Conv-TasNet, DCUnet) vs. multiple microphones (FaSNet, AFnet) as well as time domain (Conv-TasNet, FaSNet) vs. time-frequency domain (DCUnet, AFnet) processing.

---

> > ### Comment · Reviewer_aM96 · 2022-12-01
> > **Final comments**
> >
> > I thank the authors for answering my queries, however, my overall assessment of the paper still remains the same.
> > The novelty is below the standards of ICLR as here the expectation is towards a theoretically more solid work compared to empirical works at other venues.
> >
> > - My final recommendation is to consider resubmitting the work to another venue where most submissions are application based.
> >
> > - General comment: Conventional knowledge from Signal processing can be a basis for explainability or boosting the performance of DL model, and not for designing one. E.g., MFCCs are based on solid signal processing, and production/perception theory but are now  replaced with log-energies (just Fourier transform) or raw-waveforms in SOTA ASR systems like Wav2Vec. Most signal processing techniques have limitations due to underlying assumptions on the system being modeled and were developed in era of low computing.
> >
> > - 'learning to align and learning to filter leads to a more efficient and interpretable enhancement process.'
> > While explicit interpretability is missing from the paper, the one authors refer to comes from existing signal processing literature. Just citing literature is not enough as I said earlier, one can design a joint network and use signal processing for interpretability by decomposing the underlying mathematical operations.
> >
> > - 'Typical one-stage SE networks do not explicitly learn the spatial features
> > again such statements are not backed by any citations and are not generally true, at least not theoretically. In speech/audio literature, researchers have studied in-depth ways to capture spectral, spatial, and temporal features as a part of network layer/architecture.
> >
> > - 'most existing SE systems leverage empirical deep learning techniques to directly learn the mono target speech audio without thoughtfully considering the signal spatial properties within multiple microphones.'
> > This is not true e.g.,
> > Insights into Deep Non-linear Filters for Improved Multi-channel Speech Enhancement by Kristina
> > MULTI-MICROPHONE COMPLEX SPECTRAL MAPPING FOR SPEECH DEREVERBERATION by Zhong
> > Again not going into details about what existing works have done, but rather highlight the fact that existing works have used spatial information in some form for many speech applications.
> >
> > - The link between speech production/perception.......
> > There is still no theoretical backing for the same. Human perception works in many mysterious ways, and we often model the same under constraints. Multiple microphones can provide national acoustic cues but can't mimic the perception hence unless an underlying model is being employed, such statements should be avoided.
> >
> > - STFT vs. Raw-waveforms
> > My comment was not about using one domain over another but instead about working with complex numbers. Just like authors favor a split approach over a joint network as they argue it is difficult to optimize/learn important spatial information, in a similar manner working with complex networks and being able to scale them or optimize over large corpora is difficult and in my personal experience usually doesn't work.
> >
> > - MSE loss. the effect on the loss functions is beyond the scope of this paper.
> > As I said earlier, MSE or any of its variants usually don't outperform losses like STOI.
> > The impact of loss function is very important because it also impact the kind of features one learn and if at all one can learn that e.g., MSE may be a poor choice for a joint model/training. Work in Braun-21 is actually for RNN type of networks.
> >
> > - Thanks for adding more details on implementation, but these mostly cite existing implementations. An end-to-end demo implementation of what actually is (and how) implemented is very necessary for DL papers in top venues like ICLR.

---

> > > ### Author Response · Authors · 2022-12-02
> > > **Thank You for Further Comments (1/3)**
> > >
> > > We thank the reviewer for the further comments. We clarify a few more points below.
> > >
> > > > 'Typical one-stage SE networks do not explicitly learn the spatial features again such statements are not backed by any citations and are not generally true, at least not theoretically. In speech/audio literature, researchers have studied in-depth ways to capture spectral, spatial, and temporal features as a part of network layer/architecture.
> > >
> > > To explicitly capture spatial features many existing methods use specialized design of the network layer and architecture. Our approach does not require specialized design of the network building blocks as the RTF supervision is a general concept that can potentially be applied to a broader class of architectures. The related works either utilize specialized beamformer units such as MVDR in the network design for capturing spatial features, or extracting spatial information prior to feeding data into the network (such as IPDs). Our approach adopts RTF information to supervise the network to learn spatial features and bypass the need of the additional processing steps, leading to an end-to-end network that directly takes the noisy signals as input and predicts the clean signal at the output.
> > >
> > > > 'most existing SE systems leverage empirical deep learning techniques to directly learn the mono target speech audio without thoughtfully considering the signal spatial properties within multiple microphones.' This is not true e.g., Insights into Deep Non-linear Filters for Improved Multi-channel Speech Enhancement by Kristina MULTI-MICROPHONE COMPLEX SPECTRAL MAPPING FOR SPEECH DEREVERBERATION by Zhong Again not going into details about what existing works have done, but rather highlight the fact that existing works have used spatial information in some form for many speech applications.
> > >
> > > Thank you for mentioning these works. These methods either utilize conventional beamformer units (e.g., MVDR) to explicitly learn spatial features or directly learn the mono clean audio output. To be more specific, the work *“Insights into deep non-linear filters for improved multi-channel speech enhancement”* by Tesch and Gerkmann belongs to the type of existing methods discussed at _(1) Comparison to deep learning approaches based on directly learning to map multichannel signals to the mono target speech signal (see [Response to all reviewers](https://openreview.net/forum?id=DQou0RiwkR0&noteId=x3VOtYNQY07)). However, without utilizing any beamforming units within the model it is hard to interpret what spatial features the network has actually learned from the multichannel data. Our simulation results in Figure 2 of the revised manuscript have compared the performance of networks with and without explicitly incorporating spatial information (i.e., RTF supervision) and showed that spatial information cannot be trivially learned if not explicitly supervised.
> > >
> > > The work *“Multi-microphone complex spectral mapping for speech dereverberation”* by Wang and Wang belongs to _2) Comparison to conventional beamformer-aided deep learning-based approaches_ (see [Response to all reviewers](https://openreview.net/forum?id=DQou0RiwkR0&noteId=x3VOtYNQY07)). As discussed earlier, the MVDR module would require matrix inversion usually could lead to high computational complexity and numerical instability. The computation for the inverse matrix also prevents this type of methods from scaling up to systems with more microphones.

---

> > > > ### Author Response · Authors · 2022-12-02
> > > > **Thank You for Further Comments (2/3)**
> > > >
> > > > > STFT vs. Raw-waveforms My comment was not about using one domain over another but instead about working with complex numbers. Just like authors favor a split approach over a joint network as they argue it is difficult to optimize/learn important spatial information, in a similar manner working with complex networks and being able to scale them or optimize over large corpora is difficult and in my personal experience usually doesn't work.
> > > >
> > > > Thank you for sharing the experience on real-valued vs. complex-valued networks. Complex-valued networks have been widely adopted in audio processing tasks as a natural choice for processing complex-valued entities like Fourier transformed signals and wireless signals (e.g., [Hiorse, 2013]). Complex-valued networks are a generalization of real-valued networks, as real-valued networks can be viewed as a special case of complex-valued networks using zero imaginary parts.  Thus, complex-valued networks can be naturally applied to many signal processing tasks. The benefits of complex-valued network are also further studied in [Trabelsi et al., 2018] where it states:
> > > >
> > > > *“The role of representations based on complex numbers has started to receive increased attention, due to their potential to enable easier optimization (Nitta, 2002), better generalization characteristics (Hirose and Yoshida, 2012), faster learning (Arjovsky et al., 2015; Danihelka et al., 2016; Wisdom et al., 2016) and to allow for noise-robust memory mechanisms (Danihelka et al., 2016). Wisdom et al. (2016) and Arjovsky et al. (2015) show that using complex numbers in recurrent neural networks (RNNs) allows the network to have a richer representational capacity.”*
> > > >
> > > > In the STFT domain SE where the complex number’s phase component plays an important role, we believe complex-valued networks are the right choice for performing such tasks. The paper [Hu et al., 2020] describes how complex-valued networks can benefit complex ratio mask to achieve the best speech enhancement. In our study, we have also experimented with real-valued counterpart of the model and have found that complex-valued network performs better, which is in consistent with the observations.
> > > >
> > > > + [Hiorse, 2013] Hirose, _Complex-Valued Neural Networks: Theory and Applications_, John Wiley & Sons, Inc.: Hoboken, NJ, 2013.
> > > >
> > > > + [Trabelsi et al., 2018] Trabelsi et al., “Deep complex networks,” in _International Conference on Learning Representations (ICLR)_, 2018.
> > > >
> > > > + [Hu et al., 2020] Hu et al., “DCCRN: Deep complex convolution recurrent network for phase-aware speech enhancement,” in _Annual Conference of the International Speech Communication Association (Interspeech)_, 2020.

---

> > > > > ### Author Response · Authors · 2022-12-02
> > > > > **Thank You for Further Comments (3/3)**
> > > > >
> > > > > > MSE loss. the effect on the loss functions is beyond the scope of this paper. As I said earlier, MSE or any of its variants usually don't outperform losses like STOI. The impact of loss function is very important because it also impact the kind of features one learn and if at all one can learn that e.g., MSE may be a poor choice for a joint model/training. Work in Braun-21 is actually for RNN type of networks.
> > > > >
> > > > > We used the same loss function (i.e., the combined power-law compressed MSE loss) for all the models we trained in our experiments as for fair comparison to observe the key factor we aim to study, i.e., the RTF alignment aspect. Though the purpose of our work is not to compare STOI-based loss vs. MSE-type loss, we believe the comparison of the two types of loss functions is interesting. We have thus digged out some previous works regarding this comparison. Interestingly, we have found several works that have observed better performance of MSE loss over STOI loss. So the statement made by the reviewer _“MSE or any of its variants usually don't outperform losses like STOI”_ is not necessarily true. The following examples support the use of MSE-type losses, e.g., [Kolbæk et al., 2019] states:
> > > > >
> > > > > *“Our results are in line with recent empirical work and might explain the somewhat surprising result in [23]–[26], where none or only very modest improvements in STOI were achieved with STOI optimal DNNs compared to MSE optimal DNNs.”*
> > > > >
> > > > > We think that the argument on the optimal loss function for SE is still an open question. We are also interested in knowing if the reviewer has any works in mind arguing that STOI loss is better than MSE loss for SE. In our perspective and from the related works, MSE-type losses are a proper choice for SE studies.
> > > > >
> > > > > + [Kolbæk et al., 2019] Kolbæk et al., “On the relationship between short-time objective intelligibility and short-time spectral-amplitude mean-square error for speech enhancement,” in _IEEE/ACM Transactions on Audio, Speech, and Language Processing_, 2019.
> > > > >
> > > > > References in [Kolbæk et al., 2019]:
> > > > >
> > > > > + [23] M. Kolbæk, Z.-H. Tan, and J. Jensen, “Monaural speech enhancement using deep neural networks by maximizing a short-time objective intelligibility measure,” in _Proc. Int. Conf. Acoust., Speech, Signal Process._, 2018, pp. 5059–5063.
> > > > >
> > > > > + [24] Y. Zhao, B. Xu, R. Giri, and T. Zhang, “Perceptually guided speech enhancement using deep neural networks,” in _Proc. Int. Conf. Acoust., Speech, Signal Process._, 2018, pp. 5074–5078.
> > > > >
> > > > > + [25] H. Zhang, X. Zhang, and G. Gao, “Training supervised speech separation system to improve STOI and PESQ directly,” in _Proc. Int. Conf. Acoust., Speech, Signal Process._, 2018, pp. 5374–5378.
> > > > >
> > > > > + [26] S. W. Fu, T. W. Wang, Y. Tsao, X. Lu, and H. Kawai, “End-to-end waveform utterance enhancement for direct evaluation metrics optimization by fully convolutional neural networks,” _IEEE/ACM Trans. Audio, Speech, Lang. Process._, vol. 26, no. 9, pp. 570–1584, Sep. 2018.

---

> ### Author Response · Authors · 2022-11-30
> **Available for Discussion**
>
> We hope our responses have addressed the reviewer's concerns, but if not we are available/open to further address any outstanding issues.

---

> ### Author Response · Authors · 2022-12-06
> **Additional experiments and results available**
>
> Dear Reviewer aM96,
>
> Based on the suggestions and feedback from another reviewer, we have conducted further experiments and gathered more evidence to support our proposed AFnet method. We think you might be interested in taking a look at [here](https://openreview.net/forum?id=DQou0RiwkR0&noteId=Q-hdG5es2p9) and your feedback is welcome.
>
> Sincerely,
>
> Authors of Paper1144

---

### Official Review · Reviewer_4Sgp · 2022-11-03

**Confidence:** 5
**Correctness:** 3
**Technical Novelty And Significance:** 3
**Empirical Novelty And Significance:** 2
**Recommendation:** 6

**Clarity, Quality, Novelty And Reproducibility:**

The paper seems novel in its approach. It also reads well and seems reproducible.

**Details Of Ethics Concerns:**

There is not any ethics concern.

**Strength And Weaknesses:**

Strengths:
1. The idea seems novel.
2. It seems the two-stage model could work well by dividing the beamformer calculation into two separate stages and having individual targets (RTFs) for filters of the first stage.
3. It is nice that the method seems to be able to handle time-varying environments, but it has not been tested on such environments.

Weaknesses:
1. The experimentation is only on simulated data. Would the method work well for real data?
2. Comparison with another sequential multichannel method [Wang 2021] would have been nice. In this method, a beamformed signal based on an initial neural estimate is fed into a second stage (and even more stages) which refines the neural network output, hence the model makes use of spatial information implicitly.
3. The burden on the second stage is still high since it still needs to do some spectral denoising as well as spatial denoising.
4. The training uses a single room configuration which is limited. Does it generalize to unseen rooms?
5. U-net seems better than W-net itself (for mics > 2) in Figure 2. What if we used U-nets in AFNet instead of W-nets? It has been observed that using complex weights in a network is usually not beneficial, and it is better to use real valued weights within the neural network.

[Wang 2021] Wang, Zhong-Qiu, et al. "Sequential multi-frame neural beamforming for speech separation and enhancement." 2021 IEEE Spoken Language Technology Workshop (SLT). IEEE, 2021.

**Summary Of The Paper:**

The paper introduces a two-stage model for multi-channel speech enhancement. The first stage predicts STFT domain filters for each microphone that will take them close to a reference microphone's clean signal, by estimating time-varying RTFs. This is called an alignment estimator. The second stage predicts further filters for each channel and filtered channels are summed up to obtain the estimated signal which needs to be close the target signal for the reference microphone. This second step can be seen as a typical deep beamforming network.

The model for each stage is a W-net model which is like a U-net model but uses complex matrix multiplications and convolutions.

**Summary Of The Review:**

The paper is nice and innovative but it has some flaws in experimentation as listed in the weaknesses above.

Some specific issues are highlighted below:

1. In Figure 1, one of the align filters are always all ones (the one corresponding to the reference mic) and does not need to be estimated. It should be noted.
2. In Section 3, the discussion assumes a time-varying single frame STFT domain filtering to describe room impulse response filtering. However, typically RIRs are much longer than a single frame and we may assume they are time-invariant if there is no motion.
3.  Instead of SSNR, maybe report SI-SNR which is more commonly used. If using SSNR, maybe mention the segment length.
4. In Section 4.2.3, it seems obvious that alignment masks would give more spatially diverse coefficients when there is an RTF loss on them. When there is no such loss, it is not clear what those initial filters mean and how they contribute to overall system.

---

> ### Author Response · Authors · 2022-11-17
> **Response to Reviewer 4Sgp (1/2)**
>
> We would like to thank Reviewer 4Sgp for the thoughtful comments and efforts towards improving our manuscript. Hope our following replies help answer the reviewer's outstanding questions.
>
> > **It is nice that the method seems to be able to handle time-varying environments, but it has not been tested on such environments.**
>
> We thank the reviewer for suggesting that we test on time-varying room impulse response (RIR) environments. We have conducted additional experiments on evaluating the proposed AFnet on such environments and showed that it generalizes to scenarios with changing target speech location even though it has not seen such cases during the training stage. The key is that we have utilized RTF supervision on learning the alignment masks which makes it robust to various speech positions. For details, please refer to *Appendix C.6: SE under time-varying RIR scenarios* of the revised manuscript.
>
> > **The experimentation is only on simulated data. Would the method work well for real data?**
>
> In addition to the simulated multichannel data generated by using the Pyroomacoustics simulator, we also conducted experiments on more realistic data generated using real-world measured RIRs taken from [Hadad et al., 2014] as well as on real multichannel recorded data of the CHiME-3 dataset. The related discussion in *Appendix C.7: Results on real-world measured RIR generated data* of the revised manuscript shows that our observation of the proposed RTF alignment scheme is consistent in both simulated and real-world scenarios.
>
> + [Hadad et al., 2014] E. Hadad et al., “Multichannel audio database in various acoustic environments,” _in International Workshop on Acoustic Signal Enhancement (IWAENC)_, 2014
>
> > **Comparison with another sequential multichannel method [Wang 2021] would have been nice. In this method, a beamformed signal based on an initial neural estimate is fed into a second stage (and even more stages) which refines the neural network output, hence the model makes use of spatial information implicitly.**
> > + **[Wang 2021] Wang, Zhong-Qiu, et al. "Sequential multi-frame neural beamforming for speech separation and enhancement." 2021 IEEE Spoken Language Technology Workshop (SLT). IEEE, 2021.**
>
> Thank you for recommending this sequential multichannel work to be compared with. The method in the work also adopts sequential refinement to perform SE but utilizing a different way of connecting the stages. Their method utilizes multichannel Wiener filter (MWF) blocks to joint the stages together. Due to leveraging MWF, spatial information can be implicitly incorporated. However, matrix inversion is needed to compute the beamformer weights through MWF which could significantly increase the computational cost when the number of microphones increases, thus is less friendly to scale up. On the contrary, our approach joints the two stages by introducing intermediate masks via simple skip connections with RTF supervision to incorporate spatial information explicitly. We have added related discussion in the revised manuscript, *Section 3.4: Differentiation from existing methods*.
>
> > **The burden on the second stage is still high since it still needs to do some spectral denoising as well as spatial denoising.**
>
> After training the Align Net (first stage) for the RTF loss, the entire AFnet (i.e., both Align Net and Filter Net) parameters are jointly trained for the signal reconstruction loss to perform the final denoising task. The new results in *Appendix C.2: Fixing vs. unfixing Align Net while minimizing reconstruction loss* (also shown in the table below)  demonstrate that joint optimization of both stages for signal reconstruction loss (after the Aligne Net has been trained for the RTF loss) is crucial for obtaining better performance. This indicates that both stages may share the task of performing spatial and spectral denoising jointly, where the first stage shares more loading on the spatial enhancement part while the second stage is more on the spectral denoising portion. We have shown that dividing the SE task into the two subtasks makes the network learn to more efficiently denoise the speech.
>
> Table: Results of AFnet trained with fixed and unfixed Align Net after the Align Net has been trained with the RTF loss (taken from Table 6 of the revised manuscript).
> | | PESQ | | STOI |        | SSNR |      |
> |-------------------------|--------------------------|--------------------------|--------------------------|--------------------------|--------------------------|--------------------------|
> | \# Mic | Fixed | Unfixed   | Fixed | Unfixed    | Fixed | Unfixed
> | 2                       | 1.77                     | 1.84    | 0.712 | 0.728 |4.93 | 5.06
> | 4                       | 1.81                     | 1.92    | 0.721 | 0.739 | 5.01 | 5.26
> | 8                       | 1.84                     | 1.99    | 0.728 | 0.753 | 4.99 | 5.35

---

> > ### Author Response · Authors · 2022-11-17
> > **Response to Reviewer 4Sgp (2/2)**
> >
> >
> > > **The training uses a single room configuration which is limited. Does it generalize to unseen rooms?**
> >
> > We have conducted additional experiments on evaluating the proposed AFnet on unseen room configurations. The results in *Appendix C.5: Generalization to other room configurations* of the revised manuscript reveal that the proposed AFnet training is robust to unseen rooms.
> >
> > > **U-net seems better than W-net itself (for mics > 2) in Figure 2. What if we used U-nets in AFNet instead of W-nets? It has been observed that using complex weights in a network is usually not beneficial, and it is better to use real valued weights within the neural network.**
> >
> > We would like to first clarify that our AFnet already utilizes a single U-Net for each stage (i.e., a U-Net for Align Net and another U-Net for Filter Net), making the entire network a W-shaped architecture (W-Net). With regards to complex-valued operation, as our aim is to estimate complex weights for combining noisy STFTs, it is straightforward to operate the model in the complex domain to maximally leverage the complex nature of STFTs, especially when the phase component plays an essential role here. The benefit of utilizing complex operation for phase-aware SE has also been observed by the following work:
> >
> > + [Choi et al., 2018] H.-S. Choi et al., “Phase-aware speech enhancement with deep complex U-Net,” _arXiv:1903.03107_, 2018.
> >
> > > **In Figure 1, one of the align filters are always all ones (the one corresponding to the reference mic) and does not need to be estimated. It should be noted.**
> >
> > We have noted this aspect in our revised manuscript.
> >
> > > **In Section 3, the discussion assumes a time-varying single frame STFT domain filtering to describe room impulse response filtering. However, typically RIRs are much longer than a single frame and we may assume they are time-invariant if there is no motion.**
> >
> > Thank you for pointing out this underlying assumption. To encompass the broader case where the target speaker location would not always be static, we have used the _H_(f,t) for denoting the potentially time-varying acoustic transfer function. When there is no motion, we will only have _H_(f,t)=_H_(f) independent of the time axis. Our approach can be applied to time-varying cases shown in _Appendix C.6: SE under time-varying RIR scenarios_ in the revised manuscript.
> >
> >
> > > **Instead of SSNR, maybe report SI-SNR which is more commonly used. If using SSNR, maybe mention the segment length.**
> >
> > We have decided to keep using SSNR as it has been widely adopted not only in many deep learning works but also in conventional signal processing approaches. We thank the reviewer for suggesting adding the segment length information and we have done so (with more parameter settings provided) in the revised manuscript.
> >
> > > **In Section 4.2.3, it seems obvious that alignment masks would give more spatially diverse coefficients when there is an RTF loss on them. When there is no such loss, it is not clear what those initial filters mean and how they contribute to overall system.**
> >
> > Surely, when there is no RTF supervision the network itself has to figure out how the intermediate masks should be like solely by learning to predict the final mono clean target speech. Therefore, it is hard to interpret or explain what has actually been learned in the intermediate masks. Without a clear explanation of the learned variables, we admit that it will be difficult to conclude how they contribute to the system overall. However, that is really why the RTF supervision comes into play. By dividing the large network into subtasks of alignment and filtering units, the entire enhancement process of the AFnet is more interpretable as well as efficient, as the intermediate masks are associated with the spatial characteristics of the target source that it aims to extract.

---

> > > ### Comment · Reviewer_4Sgp · 2022-12-02
> > > **Response to authors' response**
> > >
> > > Thanks to the author's for responding to my comments.
> > >
> > > The additions to the paper made it a better paper, but some concerns remain. While I find the idea interesting and worth trying, I am a bit concerned with experimentation and comparisons in the paper.
> > >
> > > In terms of comparing with existing work, I think predicting multichannel filters (beamformer coefficients) through neural nets has been proposed before, so it would be good to mention them. I think the earliest such paper may be the following:
> > >
> > > [1] X. Xiao et al., "Deep beamforming networks for multi-channel speech recognition," 2016 IEEE International Conference on Acoustics, Speech and Signal Processing (ICASSP), 2016, pp. 5745-5749, doi: 10.1109/ICASSP.2016.7472778.
> > >
> > > I thought these types of works that predict filters through neural networks were missing from the discussion, so it would be good to add them.
> > >
> > > Thanks for adding experiments using measured RIRs. This is not the same as using real data, though. Chime-3 test set is also not a real test set. There is a real Chime-3 test set, but the authors are reporting and comparing results on the simulated test set of Chime-3. Real Chime-3 test data does not have references, so it is not trivial to calculate signal-level metrics (like PESQ, STOI, SSNR), whereas ASR WERs can be calculated easily on them. The paper compares with other papers' results on the simulated test-set, so I assume they also use the simulated test set.
> > >
> > > I guess a new model was trained for the measured RIRs. If a new model was trained, more details about how many training and test utterances were used should be added. Training on the same real RIRs and testing with those RIRs is still a matched condition and does not actually constitute a generalization test. If a new model was trained for this task, was there overlap between RIRs of the train and test sets. If there is an overlap, I think this fully matched condition is a very limiting condition. Even if different set of RIRs from the same room is used for train and test, still it would be quite rare in real-world use cases to always keep a device in the same room.
> > >
> > > For the unseen room tests, the dimensions for the new rooms were still similar to the original room and more importantly, it was not clear what the t60 RIR value ranges were. I believe we need more extensive room configurations and t60 ranges to see the real generalizability of this model. I assume it would be better to train from multiple room configurations and test on multiple room configurations.
> > >
> > > In general, I think the set of RIRs used for training data and set of RIRs used for testing should be distinct and different in the sense that they should be separately sampled from a range of environments and t60 values as well as locations in the room etc. It was not clear if this was the case from reading the paper.
> > >
> > > I think the paper follows the lead of some other papers to use complex network operations (complex convolutions) and claims it performs better than using real convolutions but this benefit has not been shown clearly here. Even though some earlier papers may have found complex convolutions to be better, I am aware of some other works which found that complex ops perform worse than using real ops when the input and output to the network are complex, e.g. see section V.B in [2]. That is using a real network with concatenated real-imaginary values as input and output may perform better usually, under the same parameter count constraints. This may also depend on the task and network architecture. I think the authors may be surprised to find that using real networks may end up getting better performance in their case too.
> > >
> > > [2] Williamson DS, Wang D. Time-Frequency Masking in the Complex Domain for Speech Dereverberation and Denoising. IEEE/ACM Trans Audio Speech Lang Process. 2017 Jul;25(7):1492-1501. doi: 10.1109/TASLP.2017.2696307. Epub 2017 Apr 20. PMID: 30112422; PMCID: PMC6089240.
> > >
> > > In terms of comparison with methods that do neural beamforming (or neural mask-based beamforming), that is perform beamforming (MVDR or MCWF) based on an initial output of a neural network, we do not see a comparison with such approaches in the paper. The argument made by the authors is that there is matrix inversion in those algorithms which causes speed, memory and numerical issues, but those matrix inversions are not usually very large matrix inversions (MxM matrices where M is the number of microphones) and they do not actually constitute a lot of computation and it is not difficult to stabilize them using diagonal loading.
> > >
> > > Based on these observations, I am keeping my score of 6, since I still see a potential in this method and these early but limiting experimentation may still be enough to show that the method helps under certain conditions.

---

> > > > ### Comment · Reviewer_4Sgp · 2022-12-06
> > > > **Additional comment**
> > > >
> > > > I re-wrote the paragraph about neural beamforming below, correcting and updating some issues.
> > > >
> > > > In terms of comparison with methods that do neural beamforming (or neural mask-based beamforming), that is perform beamforming (MVDR or MCWF) based on an initial output of a neural network, we do not see a comparison with such approaches in the paper except on Chime-3 simulated data where the authors compare with one of the earliest studies that performed mask-based beamforming. More recent studies use multi-frame beamformers and use their output as additional input to a second stage neural net which helps improve performance. The argument made by the authors is that there is matrix inversion in those algorithms which causes speed, memory and numerical issues, but those matrix inversions are not usually very large matrix inversions (multiple MxM matrices where M is equal to or proportional to the number of microphones) and they do not actually constitute a lot of computation and it is not difficult to stabilize them using diagonal loading.

---

> > > > > ### Author Response · Authors · 2022-12-06
> > > > > **Thank you for the additional comment**
> > > > >
> > > > > We thank the reviewer for further checking our paper and the effort spent on providing more insightful suggestions. We feel our work has been greatly improved through your constructive feedbacks.

---

> > > > ### Author Response · Authors · 2022-12-06
> > > > **More experimental results to discuss the above feedbacks (1/5)**
> > > >
> > > > We greatly appreciate your time and feedbacks to help us improve our paper quality through the iterative rebuttal discussion. Below we provide further experiments to strengthen the proposed approach based on your comments.
> > > >
> > > > > In terms of comparing with existing work, I think predicting multichannel filters (beamformer coefficients) through neural nets has been proposed before, so it would be good to mention them. I think the earliest such paper may be the following:
> > > >
> > > > > [1] X. Xiao et al., "Deep beamforming networks for multi-channel speech recognition," 2016 IEEE International Conference on Acoustics, Speech and Signal Processing (ICASSP), 2016, pp. 5745-5749, doi: 10.1109/ICASSP.2016.7472778.
> > > >
> > > > > I thought these types of works that predict filters through neural networks were missing from the discussion, so it would be good to add them.
> > > >
> > > > Thank you very much for pointing out this beamforming coefficient estimation type of methods that we missed. We will make sure to include relevant discussion on such DNN-based beamforming filter prediction approaches as we improve our paper.
> > > >
> > > > > Thanks for adding experiments using measured RIRs. This is not the same as using real data, though. Chime-3 test set is also not a real test set. There is a real Chime-3 test set, but the authors are reporting and comparing results on the simulated test set of Chime-3. Real Chime-3 test data does not have references, so it is not trivial to calculate signal-level metrics (like PESQ, STOI, SSNR), whereas ASR WERs can be calculated easily on them. The paper compares with other papers' results on the simulated test-set, so I assume they also use the simulated test set.
> > > >
> > > > Thank you for mentioning to clarify the experiments regarding CHiME-3 dataset. Yes, as the reviewer pointed out, for the CHiME-3 experiments we actually used the simulated test set rather than the real test set. The simulated test set data were obtained by mixing real-world recorded speech and noise audio streams to generate noisy signals, while the real test set data were obtained by recording the noisy speech signals directly in a noisy environment. We will make it clear that we are comparing with the simulated test data with the SOTA approaches as we improve our paper.
> > > >
> > > > > I guess a new model was trained for the measured RIRs. If a new model was trained, more details about how many training and test utterances were used should be added. Training on the same real RIRs and testing with those RIRs is still a matched condition and does not actually constitute a generalization test. If a new model was trained for this task, was there overlap between RIRs of the train and test sets. If there is an overlap, I think this fully matched condition is a very limiting condition. Even if different set of RIRs from the same room is used for train and test, still it would be quite rare in real-world use cases to always keep a device in the same room.
> > > >
> > > > Thank you for pointing out these concerns. Yes, a new model was trained for the measured RIRs in this case. The numbers of training and test files are the same as those used in the Pyroomacoustic-based RIR experiments (i.e., 8308 for training and 1199 for testing using AVSpeech files). The measured RIRs used to generate the training and test audio data are from the same set though, which means it is a relatively limiting condition as you mentioned. We will make it clear in the description of the experimental settings as we further revise our work.
> > > >
> > > > To further inspect the generalization of the method, **we have conducted additional experiments to test the model performance on unseen RIRs measured by an unseen array configuration**. To be more specific, we trained the models using the 3-3-3-8-3-3-3 array RIRs and tested them on the 4-4-4-8-4-4-4 array RIRs from the [Multi-Channel Impulse Response Database](https://www.iks.rwth-aachen.de/en/research/tools-downloads/databases/multi-channel-impulse-response-database/), where the number indicates the distance between two microphones in centimeters. The results obtained by testing on the unseen array RIRs are presented below. The trend is consistent with the results observed in the seen RIR test case (Table 7 in the revised manuscript).
> > > >
> > > > Table: SE results on test data generated using unseen RIRs:
> > > > | Method  | PESQ  | STOI  | SSNR  |
> > > > |-------------------------------------------------------------|--------------------------|--------------------------|--------------------------|
> > > > | W-Net for filter weights  | 1.86  | 0.691  | 3.09  |
> > > > | AFnet w/o RTF loss  | 2.00  | 0.727  | 3.71  |
> > > > | AFnet | **2.16**  | **0.750**  | **3.91**  |
> > > > *Note: all models are of the same size (2.6M parameters)

---

> > > > > ### Author Response · Authors · 2022-12-06
> > > > > **More experimental results to discuss the above feedbacks (2/5)**
> > > > >
> > > > > > For the unseen room tests, the dimensions for the new rooms were still similar to the original room and more importantly, it was not clear what the t60 RIR value ranges were. I believe we need more extensive room configurations and t60 ranges to see the real generalizability of this model. I assume it would be better to train from multiple room configurations and test on multiple room configurations.
> > > > >
> > > > > > In general, I think the set of RIRs used for training data and set of RIRs used for testing should be distinct and different in the sense that they should be separately sampled from a range of environments and t60 values as well as locations in the room etc. It was not clear if this was the case from reading the paper.
> > > > >
> > > > > Thank you for the suggestion on testing with more complex unseen room conditions. Yes, we agree with the reviewer that **testing on a distinct set of RIRs from the training set will better indicate the model’s generalization capabilities. We have further conducted such tests** with the simulation details presented below. The results shown in the table below indicates that the proposed method generalizes well to the more complex unseen room settings, as the RTF loss helps the AFnet to achieve better SE performance and still improves as the number of microphone increases.
> > > > >
> > > > > #### Training RIRs sampled from:
> > > > >
> > > > > - Room dimensions: {(8x5x3), (7x6x4), (10x8x5), (4x6x5), (6x6x6)} (length×width×height in meters)
> > > > >
> > > > > - RT60: {0.20, 0.35, 0.50, 0.65} (second)
> > > > >
> > > > > - Locations: {1, 2, 3} meter(s) away from center of room
> > > > >
> > > > > #### Test RIRs sampled from:
> > > > >
> > > > > - Room dimensions: {(5x5x8), (4x10x6), (7x5x7), (5x4x5), (9x8x6)} (length×width×height in meters)
> > > > >
> > > > > - RT60: {0.24, 0.34, 0.44, 0.54} (second)
> > > > >
> > > > > - Locations: {0.5, 1.5, 2.5} meter(s) away from center of room
> > > > >
> > > > > Table: SE results obtained by training and testing with multiple room configurations (different sets of room configurations were sampled for training and testing).
> > > > > |  |  | PESQ | | | STOI | | | SSNR ||
> > > > > |-------------------------------------------------------|--------------------------|--------------------------|--------------------------|--------------------------|--------------------------|--------------------------|--------------------------|--------------------------|--------------------------|
> > > > > | Method  | 2-mic | 4-mic | 8-mic | 2-mic | 4-mic | 8-mic | 2-mic | 4-mic | 8-mic |
> > > > > | W-Net for filter weights  | 1.69  | 1.73  | 1.75  | 0.707 | 0.715 | 0.718 | 4.93 | 5.07 | 5.03
> > > > > | AFnet w/o RTF loss | 1.77  | 1.72  | 1.83 | 0.720 | 0.714 | 0.731 | 5.15 | 4.98 | 5.26
> > > > > | AFnet  | **1.81**  | **1.91**  | **1.96**  | **0.726** | **0.743** | **0.751** | **5.21** | **5.48** | **5.50**
> > > > > *Note: all models are of the same size (2.6M parameters)
> > > > >
> > > > > > I think the paper follows the lead of some other papers to use complex network operations (complex convolutions) and claims it performs better than using real convolutions but this benefit has not been shown clearly here. Even though some earlier papers may have found complex convolutions to be better, I am aware of some other works which found that complex ops perform worse than using real ops when the input and output to the network are complex, e.g. see section V.B in [2]. That is using a real network with concatenated real-imaginary values as input and output may perform better usually, under the same parameter count constraints. This may also depend on the task and network architecture. I think the authors may be surprised to find that using real networks may end up getting better performance in their case too.
> > > > > > + [2] Williamson DS, Wang D. Time-Frequency Masking in the Complex Domain for Speech Dereverberation and Denoising. IEEE/ACM Trans Audio Speech Lang Process. 2017 Jul;25(7):1492-1501. doi: 10.1109/TASLP.2017.2696307. Epub 2017 Apr 20. PMID: 30112422; PMCID: PMC6089240.

---

> > > > > > ### Author Response · Authors · 2022-12-06
> > > > > > **More experimental results to discuss the above feedbacks (3/5)**
> > > > > >
> > > > > > Thank you for referring the reference that shows an example of the real-valued network performs better than the complex-valued counterpart within the SE task. We have also conducted our own experiment to compare both types of networks. In this case, the real-valued counterpart of the AFnet still utilizes the same W-Net architecture but with increased number of channels in each layer to become similar size of the complex-valued one ($C_1$, $C_2$, $C_3$, $C_4$ = 34, 68, 68, 136 for the real-valued network). The results are shown below, where the complex-valued network performs better than the real-valued one. In our case, the favor for complex-valued network of the proposed approach may be due to the increased importance of phase manipulation in the multiple microphone case against the monaural SE case.
> > > > > >
> > > > > > However, as the reviewer mentioned, we have followed the lead of the previous works that observed the merits of complex-valued network on the SE task. After further studying the related references, we have also realized that several works have also observed the opposite where the real-valued network performs better by using concatenated real-imaginary values as input and output. We agree with the reviewer that this can be a case-by-case situation, where whether real-valued or complex-valued network is better may depend on the task and network architecture. Nevertheless, we would like to emphasize that our “align-and-filter” concept through RTF supervision can potentially adopt real-valued and complex-valued network operations.
> > > > > >
> > > > > > Tables: SE performance comparison of AFnet implemented with real-valued and complex-valued network operations.
> > > > > >
> > > > > > I. Seen RIRs
> > > > > > |  |  | PESQ | | | STOI | | | SSNR ||
> > > > > > |-------------------------------------------------------|--------------------------|--------------------------|--------------------------|--------------------------|--------------------------|--------------------------|--------------------------|--------------------------|--------------------------|
> > > > > > | Method  | 2-mic | 4-mic | 8-mic | 2-mic | 4-mic | 8-mic | 2-mic | 4-mic | 8-mic |
> > > > > > | AFnet (real-valued) | 1.78  | 1.89 | 1.97 | 0.713 | 0.731 | 0.743 | 4.70 | 5.17 | 5.24
> > > > > > | AFnet (complex-valued)  | **1.84**  | **1.92**  | **1.99**  | **0.728** | **0.739** | **0.753** | **5.06** | **5.26** | **5.35**
> > > > > >
> > > > > > II. Unseen RIRs
> > > > > > |  |  | PESQ | | | STOI | | | SSNR ||
> > > > > > |-------------------------------------------------------|--------------------------|--------------------------|--------------------------|--------------------------|--------------------------|--------------------------|--------------------------|--------------------------|--------------------------|
> > > > > > | Method  | 2-mic | 4-mic | 8-mic | 2-mic | 4-mic | 8-mic | 2-mic | 4-mic | 8-mic |
> > > > > > | AFnet (real-valued) | 1.82  | 1.92 | 1.99 | 0.713 | 0.732 | 0.742 | 4.87 | 5.38 | 5.41
> > > > > > | AFnet (complex-valued)  | **1.87**  | **1.95**  | **2.01**  | **0.726** | **0.740** | **0.753** | **5.25** | **5.47** | **5.55**
> > > > > >
> > > > > > III. Time-varying RIRs
> > > > > > |  |  | PESQ | | | STOI | | | SSNR ||
> > > > > > |-------------------------------------------------------|--------------------------|--------------------------|--------------------------|--------------------------|--------------------------|--------------------------|--------------------------|--------------------------|--------------------------|
> > > > > > | Method  | 2-mic | 4-mic | 8-mic | 2-mic | 4-mic | 8-mic | 2-mic | 4-mic | 8-mic |
> > > > > > | AFnet (real-valued) | 1.71  | 1.79 | 1.84 | 0.696 | 0.712 | 0.722 | 4.45 | 4.92 | 4.96
> > > > > > | AFnet (complex-valued)  | **1.77**  | **1.82**  | **1.86**  | **0.711** | **0.720** | **0.732** | **4.84** | **5.00** | **5.07**
> > > > > >
> > > > > > *Note: all models are of the same size (2.6M parameters)
> > > > > >
> > > > > > > In terms of comparison with methods that do neural beamforming (or neural mask-based beamforming), that is perform beamforming (MVDR or MCWF) based on an initial output of a neural network, we do not see a comparison with such approaches in the paper except on Chime-3 simulated data where the authors compare with one of the earliest studies that performed mask-based beamforming. More recent studies use multi-frame beamformers and use their output as additional input to a second stage neural net which helps improve performance. The argument made by the authors is that there is matrix inversion in those algorithms which causes speed, memory and numerical issues, but those matrix inversions are not usually very large matrix inversions (multiple MxM matrices where M is equal to or proportional to the number of microphones) and they do not actually constitute a lot of computation and it is not difficult to stabilize them using diagonal loading.

---

> > > > > > > ### Author Response · Authors · 2022-12-06
> > > > > > > **More experimental results to discuss the above feedbacks (4/5)**
> > > > > > >
> > > > > > > Thank you for the suggestion. When _M_ is small the complexity is not an issue for the inverse matrix but it could become more intensive for a large _M_ as the complexity scales according to $\mathcal{O}(M^2)$ or $\mathcal{O}(M^3)$, depending on whether optimization is made for the inversion computation. And this computation may have to be performed for each time-frequency bin if computing time-varying beamformer weights (though in a more stationary environment a common set of beamformer weights can be applied across all time frames which is much less expensive). In some SE works, they also mention the potential instability of the matrix inversion when jointly trained with the network, e.g., [Zhang et al., 2021a; Zhang et al., 2021b]. While we believe that MVDR is still a very popular and powerful tool for multichannel SE and several optimizations are available for faster computation and reduced instability issues, in this work we aim to explore other possibilities without using conventional beamformer operations.
> > > > > > >
> > > > > > > + [Zhang et al., 2021a] Z. Zhang et al., “ADL-MVDR: All deep learning MVDR beamformer for target speech separation,” in _IEEE International Conference on Acoustics, Speech, and Signal Processing (ICASSP),_ 2021.
> > > > > > > + [Zhang et al., 2022b] Z. Zhang et al., “Multi-channel multi-frame ADL-MVDR for target speech separation,” in _IEEE/ACM Transactions on Audio, Speech, and Language Processing_, 2021.
> > > > > > >
> > > > > > > But we agree that it is important to compare to such neural mask-based beamforming methods. **Actually, we compared with one of such methods which is Neural BF in Table 3 of the revised manuscript that utilizes the MVDR beamformer based on the predicted time-frequency masks (thanks to the reviewer for mentioning it in the “Additional comment” below)**. Here, to provide further comparison, **we have also conducted additional comparison to the ideal ratio mask (IRM) based neural beamformer utilizing the MWF**. In this case, to estimate the IRM, a U-Net of the same size with the AFnet was trained where the number of channels in each layer of U-Net was increased to $C_1$, $C_2$, $C_3$, $C_4$ = 46, 92, 92, 92 to match the AFnet size. During inference, the MWF algorithm is performed by leveraging the predicted IRM for estimating the signal and noise power spectral density (PSD) matrices. The PSD matrices are updated recursively with a forgetting factor. The test results on the Pyroomacoustic-based dataset are presented below. We can see that the AFnet achieves superior performance to the IRM-based MWF across different sizes of microphone array among all the three metrics. Note that the performance of IRM-based MWF approach also increases with the number of microphones, as it exploits multichannel data for spatial filtering. However, as the MWF relies on several modeling assumptions based on the signal and noise statistics, the performance of the IRM-based MWF may be bounded by the optimal MWF result and thus not able to achieve as good performance as the end-to-end approach of AFnet.
> > > > > > >
> > > > > > > Tables: SE performance comparison of AFnet with time-frequency mask-based neural beamforming algorithm.
> > > > > > >
> > > > > > > I. Seen RIRs
> > > > > > > |  |  | PESQ | | | STOI | | | SSNR ||
> > > > > > > |-------------------------------------------------------|--------------------------|--------------------------|--------------------------|--------------------------|--------------------------|--------------------------|--------------------------|--------------------------|--------------------------|
> > > > > > > | Method  | 2-mic | 4-mic | 8-mic | 2-mic | 4-mic | 8-mic | 2-mic | 4-mic | 8-mic |
> > > > > > > | IRM-based MWF | 1.52  | 1.57  | 1.58 | 0.655 | 0.669 | 0.676 | 3.52 | 3.72 | 3.80
> > > > > > > | AFnet  | **1.84**  | **1.92**  | **1.99**  | **0.728** | **0.739** | **0.753** | **5.06** | **5.26** | **5.35**
> > > > > > >
> > > > > > > II. Unseen RIRs
> > > > > > > |  |  | PESQ | | | STOI | | | SSNR ||
> > > > > > > |-------------------------------------------------------|--------------------------|--------------------------|--------------------------|--------------------------|--------------------------|--------------------------|--------------------------|--------------------------|--------------------------|
> > > > > > > | Method  | 2-mic | 4-mic | 8-mic | 2-mic | 4-mic | 8-mic | 2-mic | 4-mic | 8-mic |
> > > > > > > | IRM-based MWF | 1.55  | 1.60  | 1.62 | 0.659 | 0.674 | 0.681 | 3.82 | 4.04 | 4.13
> > > > > > > | AFnet  | **1.87**  | **1.95**  | **2.01**  | **0.726** | **0.740** | **0.753** | **5.25** | **5.47** | **5.55**

---

> > > > > > > > ### Author Response · Authors · 2022-12-06
> > > > > > > > **More experimental results to discuss the above feedbacks (5/5)**
> > > > > > > >
> > > > > > > > III. Time-varying RIRs
> > > > > > > > |  |  | PESQ | | | STOI | | | SSNR ||
> > > > > > > > |-------------------------------------------------------|--------------------------|--------------------------|--------------------------|--------------------------|--------------------------|--------------------------|--------------------------|--------------------------|--------------------------|
> > > > > > > > | Method  | 2-mic | 4-mic | 8-mic | 2-mic | 4-mic | 8-mic | 2-mic | 4-mic | 8-mic |
> > > > > > > > | IRM-based MWF | 1.54  | 1.59  | 1.61 | 0.649 | 0.663 | 0.670 | 3.48 | 3.67 | 3.77
> > > > > > > > | AFnet  | **1.77**  | **1.82**  | **1.86**  | **0.711** | **0.720** | **0.732** | **4.84** | **5.00** | **5.07**
> > > > > > > >
> > > > > > > > *Note: all models are of the same size (2.6M parameters)
> > > > > > > >
> > > > > > > > > Based on these observations, I am keeping my score of 6, since I still see a potential in this method and these early but limiting experimentation may still be enough to show that the method helps under certain conditions.
> > > > > > > >
> > > > > > > > Thank you again for the positive feedback on our approach. More importantly, **we are grateful for the valuable suggestions that have guided us to conduct the additional experiments presented above** which help further improve our work. We hope you find them also interesting; and, reinforce your positive view of the paper and make you more supportive of our work.

---

> ### Author Response · Authors · 2022-11-30
> **Available for Discussion**
>
> We hope our responses have addressed the reviewer's concerns, but if not we are available/open to further address any outstanding issues.

---

### Author Response · Authors · 2022-11-17
**Response to all reviewers (1/2)**

We would like to thank all the reviewers for their valuable time, efforts, and detailed comments dedicated to our work that will most certainly help us improve the paper, given the opportunity. Below we illustrate the motivation of our work and the difference with existing approaches, followed by outlining the major revisions made after digesting the comments from the reviewers.

## Differentiation with existing multichannel speech enhancement (SE) works:

Our system is deeply motivated by the “align” and then “filter” design principle which has been well adopted in many signal processing-based multichannel audio processing systems. Specifically, phase alignment can usually boost SNR through constructive combination of the target source signals by compensating their time delays while attenuating interferences. The phase aligned signals can be leveraged for the subsequent filtering process, leading to better denoising performance [Trees, 2004].

Although several existing deep learning frameworks [Wang et al., 2021; Zhang et al., 2021; Koyama & Raj, 2019; Koyama & Raj, 2020; Li et al., 2022] have utilized two-stage neural network designs, they still have some shortcomings that prevent their systems from fully exploiting the spatial property enabled by multiple microphones. Till now, existing two-stage deep networks that perform spatial filtering or beamforming can be considered either (1) conventional beamformer-aided approaches or (2) directly learning to map multichannel signals to the mono target speech signal. **Both still are not fully tackling the “alignment” problem and we describe each of their shortcomings below.** From our perspective, we discover that **RTF supervision for explicit incorporation of spatial features** is a major missing component for a **concise but powerful design** of multichannel SE.

### *(1) Comparison to conventional beamformer-aided deep learning-based approaches [Wang et al., 2021; Zhang et al., 2021]:*

This type of works leverages conventional beamformer units such as MVDR to bridge the network modules for learning spatial features implicitly. Such approaches often require matrix inversion or eigendecomposition operations for obtaining meaningful spatial features. However, matrix inversion usually could lead to high computational complexity and numerical instability, as the inverse matrix may not exist in the case of highly correlated signal and noise subspaces. The computation for the inverse matrix also prevents this type of methods from scaling up to systems with more microphones. To avoid so, one way is to replace such operations by using additional dedicated network layers. However, this could result in a large network which is often not desirable in audio processing applications.

**Different from these works, our method introduces a simple skip connection of alignment masks trained via supervised learning of the RTFs to explicitly incorporate interpretable spatial characteristics without the need of any beamformer unit (and thus complicated matrix operations such as matrix inversion) for learning useful spatial features.**

### *(2) Comparison to deep learning approaches based on directly learning to map multichannel signals to the mono target speech signal [Koyama & Raj, 2019; Koyama & Raj, 2020; Li et al., 2022]:*

These methods concatenate two DNN modules and jointly train both to directly learn the final **mono clean speech signal.** However, without utilizing any beamforming units within the model it is hard to interpret what spatial features the network has actually learned from the multichannel data. In other words, blindly training a two-stage network to directly predict the final mono clean speech may not fully exploit the spatial information, as the model learning process is solely driven by the mono output audio and therefore not guaranteed to perform proper spatial alignment anywhere. Consequently, the performance may not necessarily improve when the number of microphones increases as the network learning only relies on the single-channel output information.

**On the other hand, our proposed RTF supervision learning approach guides the network to compensate for the temporal delays and level differences of the microphone channels before fed to the second stage. Thus, with all the microphone channels being explicitly supervised for alignment, the neural network can better learn how to enhance the signal by exploiting all channel spatial characteristics simultaneously rather than driven solely by a mono audio output.**

---

> ### Author Response · Authors · 2022-11-17
> **Response to all reviewers (2/2)**
>
>
> ### *(3) Other differentiation points of our AFnet with related works:*
>
> **a) Align-then-filter process mimicking human listening behavior.**
> Aligning the microphone array (Align Net) acts similarly to the human listening behavior of facing toward the speaker for better perception of the speech. Therefore, an efficient SE system can first align its microphone signals in response to the target speech, followed by spectral processing for fine-tuning and enhancement. Existing deep learning approaches have yet to be designed based on such observations.
>
> **b) Complex-valued network operations.**
> Our AFnet adopts complex-valued operations of the network parameters while the existing multichannel SE works only utilize real-valued networks. The complex nature of STFTs and the alignment and filtering masks makes the complex-valued network a suitable choice for maximally exploiting spatial and spectral features of the speech in the time-frequency domain.
>
> **c) Two-stage MIMO-MIMO system of the W-Net.**
> The existing two-stage W-Net like systems utilize MIMO-MISO [Li et al., 2022] or MISO-MIMO [Koyama & Raj, 2019; Koyama & Raj, 2020] architectures (indicating the types of the two U-Net modules). Different from their works, our AFnet is a MIMO-MIMO system where both U-Nets take multichannel signals as input and estimate multichannel signals (masks) at the output. This way, the multichannel information will be better preserved from the input through the output side before later combination for the enhancement outcome.
>
> + [Trees, 2004] H. L. Van Trees, *Optimum Array Processing: Part IV of Detection, Estimation, and Modulation Theory, Honoken,* NJ, USA: John Wiley & Sons, 2004.
>
> + [Wang et al., 2021] Z.-Q. Wang et al. "Sequential multi-frame neural beamforming for speech separation and enhancement," in *IEEE Spoken Language Technology Workshop (SLT)*, 2021.
>
> + [Zhang et al., 2021] Z. Zhang et al., “ADL-MVDR: All deep learning MVDR beamformer for target speech separation,” in *IEEE International Conference on Acoustics, Speech, and Signal Processing (ICASSP),* 2021.
>
> + [Koyama & Raj, 2019] Y. Koyama and B. Raj, “W-Net BF: DNN-based beamformer using joint training approach,” *arXiv:1910.14262,* 2019.
>
> + [Koyama & Raj, 2020] Y. Koyama and B. Raj, “Exploring optimal DNN architecture for end-to-end beamformers based on time-frequency references,” *arXiv:2005.12683,* 2020.
>
> + [Li et al., 2022] J. Li et al., “The PCG-AIID system for L3DAS22 Challenge: MIMO and MISO convolutional recurrent network for Multi channel speech enhancement and speech recognition” in *IEEE International Conference on Acoustics, Speech, and Signal Processing (ICASSP),* 2022.
>
> ## Outline of revisions:
>
> 1. Modified the abstract and introduction to make sure the above-mentioned contributions of utilizing RTF information within the two-stage, sequential masking design of AFnet are well conveyed.
>
> 2. Provided more evidence to demonstrate the proposed RTF supervision of the alignment mask learning is the key to achieve remarkable improvement for multichannel SE where the target speech may come from arbitrary, time-varying locations. New results and discussion have been included to the revised manuscript at:
>
> - *Section 3.4: Differentiation from existing methods*
>
> - *Section 4.2.6: Are intermediate alignment masks necessary?*
>
> - *Appendix A: Conventional signal processing beamforming algorithms based on align-then-filter process*
>
> - *Appendix C.1: Comparison of RTF-aware training schemes (two-step vs. regularized)*
>
> - *Appendix C.2: Fixing vs. unfixing Align Net while minimizing reconstruction loss*
>
> - *Appendix C.5: Generalization to other room configurations*
>
> - *Appendix C.6: SE under time-varying RIR scenarios*
>
> 3. Changed several inline equations to be stand-alone for better readability. Modified the math notations by using bold, capital, non-italic letters for matrices and normal, capital, italic letters for their entries following the signal processing convention to improve the readability of the equations describing the proposed approach.
>
> 4. Added supervised learning details in the AFnet figure (Figure 1) to better describe the proposed align-and-filter concept along with Algorithm 1 in *Appendix B.1: RTF-aware training algorithms* of the revised manuscript.

---

### Decision · Program_Chairs · 2023-01-20

**Decision:**

Reject

**Justification For Why Not Higher Score:**

Limited novelty, missing experiments and marginal improvements, as was discussed in the meta review.

**Justification For Why Not Lower Score:**

N/A

**Metareview: Summary, Strengths And Weaknesses:**

The authors present an algorithm for multichannel speech enhancement. The algorithm, which the authors call align-and-filter net, is a 2-stage system, wherein, the first stage computes an approximate relative transfer function (RTF) between the multiple microphones with respect to a reference microphone, with the goal of aligning the multiple channels. The 2nd stage computes the enhancement filters. The first stage is trained specifically to estimate RTFs, and the second stage on enhancement. Results show gains on simulated datasets, but smaller gains on more realistic sets like Chime3, when comparing models trained w/ and w/o RTF loss.

The reviewers agreed that the idea of using 2 stages, with the first stage specifically focusing on RTF estimation is novel. The authors also present a range of results (including in appendix) that show consistent, but marginal gains compared to a similar model without RTF loss (the closest baseline, since they are trained in similar settings).

The reviewers also pointed out prior works in this space. There are multistage approaches for enhancement in the literature (although not in the same form), which the authors have not sufficiently compared against. The authors also do not compare against more recent techniques that use beamforming together with neural nets (like predicting a mask for covariance computation). In their rebuttal, the authors argue that such approaches have to compute an inverse of the matrix, which can be ill-defined. While true, such approaches are widely used in practice. The reviewers also pointed out that the size of the matrix to invert is usually not large, as they are done independently for each channel (at least when the number of microphone channels is not huge). There are also several prior approaches that aim to estimate RTFs. This dilutes the overall novelty, although the specific form in which the authors put the system together is novel. Together with the observation that the improvements using the proposed algorithm are marginal, there is consensus among reviewers that the paper does not meet the high acceptance threshold.


**Summary Of Ac-Reviewer Meeting:**

This paper was borderline, but trending towards a reject. The main discussion points during the meeting:
- The paper is reasonably well written, but novelty is limited. Furthermore, the results are not compelling enough. The authors did not directly address why they didn’t more directly compare with masking-based beamforming and other multi-stage approaches.
- The idea is interesting, since it tries to mimic the typical stages of conventional beamforming, but experimental validation is lacking.

There was consensus fairly quickly in the meeting that the paper does not meet the quality bar.